# High hedgehog signaling is transduced by a multikinase-dependent switch controlling the apico-basal distribution of the GPCR smoothened

**Marina Gonçalves Antunes, Matthieu Sanial, Vincent Contremoulins, Sandra Carvalho, Anne Plessis\*, Isabelle Becam\***

Université Paris Cité, CNRS, Institut Jacques Monod, Paris, France

**Abstract** The oncogenic G-protein-coupled receptor (GPCR) Smoothened (SMO) is a key transducer of the hedgehog (HH) morphogen, which plays an essential role in the patterning of epithelial structures. Here, we examine how HH controls SMO subcellular localization and activity in a polarized epithelium using the *Drosophila* wing imaginal disc as a model. We provide evidence that HH promotes the stabilization of SMO by switching its fate after endocytosis toward recycling. This effect involves the sequential and additive action of protein kinase A, casein kinase I, and the Fused (FU) kinase. Moreover, in the presence of very high levels of HH, the second effect of FU leads to the local enrichment of SMO in the most basal domain of the cell membrane. Together, these results link the morphogenetic effects of HH to the apico-basal distribution of SMO and provide a novel mechanism for the regulation of a GPCR.

## Editor's evaluation

In this paper, Gonçalves-Antunes and colleagues uncovered that the morphogen Hedgehog regulates the activity and subcellular localisation of Smoothened through vesicular trafficking. In particular, they demonstrated that Smoothened trafficking favours recycling and basal enrichment depends on its phosphorylation signature in a Hh-concentration-dependent manner. This work will interest a wide readership as it links Hh's functions as a morphogen with Smoothened's subcellular localisation.

**\*For correspondence:**
anne.plessis@ijm.fr (AP);
isabelle.becam@ijm.fr (IB)

**Competing interest:** The authors declare that no competing interests exist.

## Introduction

During development, signaling pathways control epithelial morphogenesis by acting on cell proliferation, differentiation, survival, and migration. Epithelial cells uniquely display an apico-basal (Ap-Ba) polarity with differential distribution of phospholipids and protein complexes between the different membrane domains (for review see *Ikenouchi, 2018*). This leads to functionally separated subregions with distinct properties and physiological functions, such as the microvilli in the apical domain, cell-cell adhesion junctions in the lateral domain, and cell-matrix adhesion in the basal domain. Finally, the apical and basal regions are in contact with extracellular environments, which can differ in the nature and dose of signaling molecules. For all these reasons, the control of signaling receptor distribution among these specific subdomains of the plasma membrane is expected to be critical for correct signal transduction.

The conserved hedgehog (HH) signals play major roles in the development of metazoans. Initially identified in the fly model, HH signaling is involved in the promotion, development, and/or metastasis

of numerous types of tumors and drugging this pathway is a major goal for cancer therapies (for review see *Ingham, 2022*). In flies, HH controls the patterning of many structures including the wing imaginal disc (WID), which has been instrumental in the study of HH signaling (for review see *Hartl and Scott, 2014*). In this epithelial structure, HH emanating from the posterior (P) cells signals to the anterior (A) cells near the A/P boundary, thus controlling the expression of target genes in a dose-dependent manner. Of note, HH molecules form two gradients in the WID: an apical gradient, required for long-range, 'low HH' responses (as in the expression of *decapentaplegic* [*dpp*]) that depend on glypicans, and a basal one, for short-distance, 'high HH' responses (as in the anterior expression of *engrailed* [*en*]), that is based on the transport of HH by exosomes associated to filopodia-like structures oriented in the A/P axis (*D'Angelo et al., 2015* and *González-Méndez et al., 2017*).

HH transduction requires Smoothened (SMO), a G-protein-coupled receptor (GPCR) like protein. In the absence of HH, the HH co-receptor Patched (PTC) inhibits SMO, probably by depleting accessible cholesterol from the outer leaflet of the plasma membrane (*Kinnebrew et al., 2021* and see for review *Radhakrishnan et al., 2020*), which promotes the formation of a repressor form of the transcription factor cubitus interruptus (CI). The binding of HH to PTC inhibits its negative effect, which blocks the cleavage of CI, leading to the transcription of pathway target genes by full-length CI. SMO acts as a scaffold to transduce HH signaling to CI via an intracellular complex (called HTC for HH Transduction Complex) bound to its cytoplasmic C-terminal domain, which includes CI and a protein kinase called Fused (FU) (*Malpel et al., 2007*; *Robbins et al., 1997*; *Sisson et al., 1997*). SMO activation is associated with conformational switches, both in its cytoplasmic C-terminal domain and in its extracellular domains; these events correlate with its clustering, which seems critical for the downstream activation of the pathway (*Fan et al., 2012*; *Shi et al., 2011*; *Su et al., 2011*; *Zhao et al., 2007*).

Several labs—including ours—have highlighted the role of endocytic trafficking and the importance of post-translational modifications in the regulation of SMO's levels, localization, and activation. SMO activation in the presence of HH is associated with changes in its localization: from internal vesicles to the plasma membrane in *Drosophila* and from the cell body to the primary cilium in mammals (*Denef et al., 2000* and *Huangfu et al., 2003*). Both events are positively controlled by extensive phosphorylation of SMO's intracellular tail by multiple kinases (for review see *Chen and Jiang, 2013*). Despite significant differences, the processes involved are remarkably conserved as illustrated by the fact that human SMO can be relocalized in response to HH to the surface of fly cells (*De Rivoyre et al., 2006*). In *Drosophila*, many kinases (protein kinase A [PKA], casein kinase I [CKI], GPCR kinase 2, casein kinase 2, Gilgamesh, atypical protein kinase C [aPKC]) are implicated, which regulate SMO activation and accumulation at the membrane (*Apionishev et al., 2005*; *Chen et al., 2010*; *Jia et al., 2010*; *Jia et al., 2004*; *Li et al., 2016*; *Maier et al., 2014* and *Zhang et al., 2004*). We have also identified a phosphorylation-based positive feedback loop between SMO and the FU kinase, which is required for the response to the highest doses of HH (*Alves et al., 1998*). In this process, the initial activation of SMO promotes the recruitment at the cell membrane of FU and its activation, which then further phosphorylates SMO, leading to an enhanced accumulation of the SMO/FU complex at the cell surface and high signaling activation (*Claret et al., 2007*; *Sanial et al., 2017*).

Are these events polarized along the Ap-Ba axis and what is their link with the gradients of HH? Previous studies indicated that SMO is unevenly distributed along the Ap-Ba axis of the WID epithelial cells (*Denef et al., 2000*; *Jiang et al., 2014*; *Sanial et al., 2017*). Here, by specifically labeling the population of SMO at the plasma membrane, we show that it is unevenly distributed along the Ap-Ba axis and that HH acts in a dose-dependent manner to increase its accumulation in the most basal region. Blocking the endocytosis of SMO or following its fate after endocytosis, reveal that SMO is initially targeted to the apical membrane and that HH does not dramatically affect its apical endocytosis but affects its post-endocytic fate, favoring recycling over degradation. Moreover, we provide evidence that the HH-dependent basolateral enrichment of SMO relies on a two-step action of the FU kinase, first apically to enhance SMO localization at the cell surface before stabilizing it in the basolateral region. Altogether, these data support a model which connects the HH-dose-dependent activation of SMO to its vesicular trafficking and Ap-Ba localization.

## Results

## High levels of HH promote a basolateral enrichment of cell surface SMO

As HH promotes the accumulation of SMO at the plasma membrane, we set out to determine (i) whether the population of SMO that is present at the cell surface is differentially distributed along the Ap-Ba axis and (ii) whether its Ap-Ba distribution is affected by HH. For that purpose, a fusion between the extracellular N-terminus of SMO and the enzymatic self-labeling SNAP-tag (*Tirat et al., 2006*), called SNAP-SMO, (*Sanial et al., 2017*) was overexpressed in the dorsal compartment of the WID (see *Figure 1A–A'* for the organization of this disc). This fusion was fully functional as its expression under *smo*'s endogenous promotor (from a BAC construct) rescued a loss of *smo* function (*Figure 1—figure supplement 1A, B-B"*). Moreover, its overexpression has no effect on HH signaling (*Figure 1—figure supplement 1C-C"*, see also *Sanial et al., 2017*). After dissection, the fraction of SNAP-SMO present at the cell surface (thereby called Surf SNAP-SMO) was specifically labeled using a non-liposoluble fluorescent SNAP ligand (*Figure 1—figure supplement 2A, A'*) before being imaged.

We imaged XY sections of wing discs labeled for SNAP-SMO at different positions along the Ap-Ba axis (*Figure 1A, B, B' and B"*). As expected, Surf SNAP-SMO levels are lower in the more A cells (called thereby far anterior, FA) away from the HH source than in the A cells abutting the A/P boundary (and that respond to HH) or in the P cells (where HH signaling is activated due to the lack of *ptc* expression) (*Méthot and Basler, 1999*). Strikingly, the increased accumulation of SNAP-SMO in the cells in which HH signaling is activated is particularly visible in the lateral section (*Figure 1B'*) and even more in the basal section (*Figure 1B"*) than in the apical region of the cells (*Figure 1B*). This basolateral enrichment is also highlighted in the reconstituted antero-posterior XZ sections (*Figure 1A', C and C"*).

To quantify the effects of HH on Surf SNAP-SMO levels and distribution, we measured (using XZ projections of eight sections) both its mean intensity (sum of pixel values over the number of pixels) and the integrated density (sum of pixel values) in three regions along the Ap-Ba axis (*Figure 1—figure supplement 2C*): (i) the apical region, estimated here as the 15% most apical region based on Discs large (DLG) staining of the septate junctions, (ii) the basal region, arbitrarily defined as being the 10% most basal part, and (iii) the lateral or intermediate region in between the two others (*Figure 1—figure supplement 2C*). Since HH acts as a morphogen in the wing disc, we performed these quantifications in four regions across the wing disc epithelium, based on the co-immunodetection of the transcription factor CI (*Figure 1C' and C"*, *Figure 1—figure supplement 2C*): the P compartment (green, where CI is not expressed) and three anterior regions: (i) the CI-R region (red, corresponding to the FA region), where CI is processed into its shorter repressor form that is not detectable with the antibody used here, (ii) the CI-F region (pink), corresponding to cells that receive medium to low levels of HH that lead to the stabilization of full-length CI, and (iii) the CI-A region (purple), which corresponds to the cells nearer to the A/P border, where CI-F is very active (and called CI-A) but present at low levels due to the repression of *ci* by the anterior expression of *en* promoted by high HH levels (*Roberto et al., 2022*). The comparison of the mean intensities of Surf SNAP-SMO in these four regions shows that HH progressively increases the levels of SNAP-SMO, with Surf SNAP-SMO being 1.7-fold more abundant in the P region than in the CI-R region (p-value) (*Figure 1D*, gray columns). These effects are seen in all the regions along the Ap-Ba axis (*Figure 1D*, the light, medium, and dark blue colors correspond to the apical, lateral, and basal regions, respectively). However, the calculation of the relative intensities of SNAP-SMO (calculated as the ratio of the integrated density of each region along the Ap-Ba axis over the integrated density of the three regions together, called column) shows that this increase in SNAP-SMO levels is unequal along the Ap-Ba axis, with a relative decrease in the apical fraction associated with a relative enrichment of the lateral and basal fractions (*Figure 1E*).

Importantly, a similar increase in the basolateral localization of SMO in the anterior region abutting the A/P and the P regions is also seen for immunolabeled endogenous SMO (*Figure 1—figure supplement 3A, A", B*), or when we labeled Surf SNAP-SMO expressed at an endogenous level from a BAC construct (*Figure 1—figure supplement 3C, C"*). Note that when we also specifically labeled the intracellular fraction of SNAP-SMO, neither its levels nor its Ap-Ba distribution was affected by HH (*Figure 1—figure supplement 2A, A"', D, D', E*).

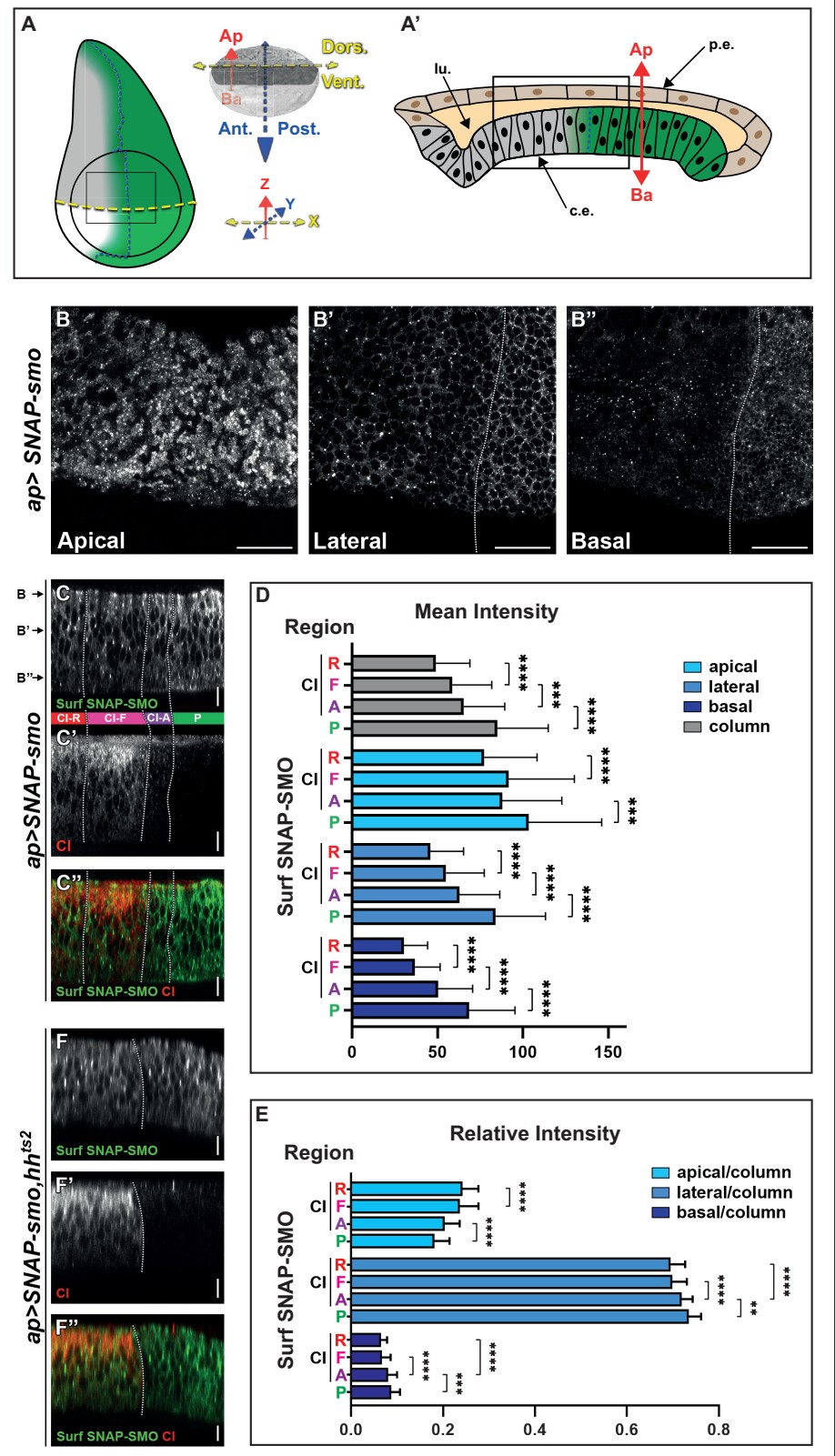

**Figure 1.** Cell surface Smoothened (SMO) is unevenly distributed along the apico-basal (Ap-Ba) axis and high levels of hedgehog (HH) promote its basolateral enrichment. (**A–A'**) (**A**) Left: scheme of a wing imaginal disc (WID) with the posterior (**P**) compartment, where HH (in green) is produced, the anterior (**A**)/P boundary is represented by the dotted dark blue line and the dorsal/ventral (D/V) boundary by the dotted yellow line. The fading green

*Figure 1 continued on next page*

*Figure 1 continued*

color represents the anterior gradient formed by HH (across about 12 rows of cells). The *apGal4* driver used to express *SNAP-smo*, is expressed in the dorsal compartment. The circle represents the wing pouch. The rectangle indicates the region that is shown in the XY images. Right: the x (D/V), y (A/P), and z (Ap-Ba) axis used for imaging are in yellow, blue, and red, respectively. Here and in all the XY images, the dorsal compartment is to the top. (**A'**) Scheme of an antero-posterior XZ section of the wing pouch with the columnar epithelium at the bottom (c.e., which is represented in (**A**) and studied here) and the peripodial epithelium at the top (p.e.), with their respective apical sides facing a lumen (lu.). (**B–B"**) Confocal XY sections along the Ap-Ba axis (as indicated) of a third instar larva WID expressing *UAS SNAP-smo* (driven by *apGal4*) in the dorsal compartment and labeled with a non-liposoluble fluorescent SNAP substrate. The scale bar represents 20 μm. Here and in the other XY images: the dorsal part of the wing pouch is shown, the discs are oriented anterior to the left, dorsal up, and the A/P boundary is represented by a vertical dotted white line drawn based on the absence of cubitus interruptus (CI) labeling in the P compartment (see *Figure 1—figure supplement 2B*). Here, in C–C" and F–F" the images were acquired using the confocal LSM980 spectral Airyscan 2, 63×. (**C–C"**) Confocal XZ antero-posterior image in the dorsal part of an *apGal4; UAS SNAP-smo* third instar larva wing pouch labeled for Surf SNAP-SMO with a non-liposoluble fluorescent SNAP substrate (C, green in C") and immunolabeled for CI-F (C', red in C"). Merged image in (**C"**). Based on CI staining, four regions (represented between the C and C' images and separated by dotted white lines) are identified along the antero-posterior axis: the P region (green), and three A regions: CI-A (low levels of full-length activated CI, in purple), CI-F (higher levels of CI full-length, in pink) and CI-R (CI repressor not detected by the anti-CI antibody used here, in red). This image was reconstituted from the XY stack that includes the three sections shown in B–B". Here and in the other XZ images: the discs are oriented anterior to the left, apical up, the A/P boundary is represented by a vertical dotted white line, and the scale bars are 10 μm. (**D–E**) Graphs showing the mean intensity (**D**) and the relative intensity (calculated for each region along the Ap-Ba axis as the ratio of its integrated density over the integrated density of the height of the disc, which is designated the column) (**E**) of Surf SNAP-SMO in the apical, lateral, and basal domains (in light, medium, and dark blue, respectively) of the CI-R (**R**), CI-F (**F**), CI-A (**A**), and P regions. N=33. Here and in the other figures, all the quantifications were done using projections of eight XZ sections that were directly acquired using an SP5 AOBS confocal, 40×. The error bars represent the SD and the statistical analysis was performed using paired t-test for the mean intensities and Wilcoxon matched-pairs signed rank tests for the relative intensities. (**F–F"**) Confocal XZ images in the dorsal part of WIDs from *apGal4; UAS SNAP-smo, hh^{ts2}* flies in which HH was inactivated at the restrictive temperature prior to dissection and labeling of Surf SNAP-SMO (F, green in F") and CI-F (indicated here as CI, in F', red in F"). Merged image in (**F"**).

The online version of this article includes the following source data and figure supplement(s) for figure 1:

**Source data 1.** Mean intensity of Surf SNAP-SMO along the apico-basal axis.

**Source data 2.** Relative intensity of Surf SNAP-SMO along the apico-basal axis.

**Figure supplement 1.** SNAP-Smoothened (SNAP-SMO) activity.

**Figure supplement 2.** Analysis of the apico-basal (Ap-Ba) distribution of SNAP-Smoothened (SNAP-SMO).

**Figure supplement 2—source data 1.** Mean intensity of Intra SNAP-SMO along the apico-basal axis.

**Figure supplement 3.** Apico-basal (Ap-Ba) distribution of immunolabeled endogenous Smoothened (SMO) or Surf SNAP-SMO expressed at endogenous levels.

**Figure supplement 3—source data 1.** Relative intensity of endogenous SMO (immunolabeling).

To ensure that these changes in the Ap-Ba distribution of Surf SNAP-SMO were indeed due to HH, we performed the same experiment when the HH function was inactivated. For that purpose, we looked at Surf SNAP-SMO in a genetic context homozygous for a thermosensitive allele of *hh* (*hh^{ts2}*) (*Ma et al., 1993*). At restrictive temperature, the function of HH is reduced, and both the anterior expression of *en* and the reduction in CI-F levels in the cells abutting the A/P boundary are suppressed (*Figure 1F' and F"*, *Figure 1—figure supplement 2F,F"*). Under that condition, Surf SNAP-SMO is no longer accumulated in the P and the anterior cells near the A/P and is no longer enriched in the basolateral region of these cells (*Figure 1F, F' and F"*).

In summary, together these data provide evidence that SMO is asymmetrically distributed along the Ap-Ba axis, and that HH acts in a dose-dependent manner on the Ap-Ba distribution of Surf SNAP-SMO, leading to an increase in the lateral and especially basal population in presence of the highest levels of HH.

## HH controls the fate of SMO post-endocytosis

To understand how the distribution of Surf SNAP-SMO along the Ap-Ba axis is established, we looked at the consequences of blocking its endocytosis.

For that purpose, we first used a thermosensitive mutation of the *shibire (shi^{ts})*, a *Drosophila* Dynamin ortholog, which is central for the scission of coated vesicles (*van der Bliek and Meyerowitz, 1991*). Blocking SHI activity—for less than an hour—leads to an accumulation of Surf SNAP-SMO in both compartments of the WID, with a stronger accumulation in the apical region of the cells (*Figure 2A, B*, *Figure 2—figure supplement 1A, B*).

Quantification of the mean intensities in the FA region (corresponding to Cl-R, no HH) and P region are shown in *Figure 2C*. It reveals that the increase in Surf SNAP-SMO levels is comparable in both regions and that it occurs all along the Ap-Ba axis of the cells, with an especially strong increase in the apical region. Calculation of the relative intensities (*Figure 2C'*) confirms the relative enrichment of Surf SNAP-SMO in the apical region of the cells and shows that it is associated with its relative decrease in the lateral region. Note that the effects seen here are specific to the inactivation of *shi^{ts}* by the restrictive temperature as (i) the distribution of Surf SNAP-SMO is similarly affected when comparing *shi^{ts}* flies at permissive and restrictive temperature (*Figure 2—figure supplement 1E*), whereas (ii) *shi^{ts}* and *shi^+* control (ctr) flies have indistinguishable distributions when kept at the permissive temperature (+).

To confirm these results, we also blocked SMO trafficking just after endocytosis (in early endosomes), using a constitutively active RAB5 tagged with a YFP (YFP-RAB5^{CA}) that is locked in the GTP bound state (*D'Angelo et al., 2015*). When YFP-RAB5^{CA} is expressed for 24 hr, Surf SNAP-SMO strongly accumulates in YFP-RAB5^{CA} positive endocytic vesicles, which are almost exclusively located in the apical region (*Figure 2D and E–E''*, *Figure 2—figure supplement 1C, D, D''*). This leads to a decrease in the relative abundance of its lateral fraction and to a lesser extent, of its basal fraction (*Figure 2F*). These effects are seen both in the presence (P region) and absence (FA region) of HH but are slightly weaker in its presence. Of note, YFP-RAB5^{CA} overexpression also led to the accumulation of endogenous SMO or SNAP-SMO expressed at endogenous levels in apical YFP-RAB5^{CA} positive endocytic vesicles (*Figure 2—figure supplement 1G-G'',"H-H'''*).

Finally, we ruled out an indirect effect of a general block of endocytosis, as we obtained similar results when we specifically blocked the endocytosis of SNAP-SMO by downregulating (by RNA interference, RNAi) the expression of *smurf*, which encodes an E3 ubiquitin ligase known to promote SMO endocytosis by mediating its ubiquitylation (*Li et al., 2018*; *Figure 2G*).

In conclusion, blocking SMO endocytosis/in early endosomes by three different means reveals that (i) SMO endocytosis is not dramatically affected by HH and (ii) SMO endocytosis occurs all along the Ap-Ba axis but more apically than basolaterally. They also show that newly synthesized SMO is initially addressed and subsequently endocytosed at the apical membrane.

## Endocytosed SMO is targeted from the apical to the basolateral region

The above data indicate that the strong stabilization of SMO in response to HH is likely due to a reduction of its degradation after endocytosis. To study the fate of Surf SNAP-SMO after its endocytosis, we performed an endocytosis assay in which Surf SNAP-SMO labeling was followed by a chase. After the chase, the subcellular localization of Surf labeled SNAP-SMO shifted: its presence at the cell surface decreases, and this is associated with increased localization in dot-like structures that likely correspond to endocytic vesicles as some of them colocalize with RAB7, an endosomal marker required for the trafficking between late endosomes and the lysosome (*Vanlandingham and Ceresa, 2009*; *Figure 3A and B* and *Figure 3—figure supplement 1A, A', A'', B, B', B''*). The global levels of Surf labeled SNAP-SMO also decrease during the chase in the whole disc with a reduction of 44% in the far anterior cells (no HH) but of only 22% in the posterior cells (with HH) (*Figure 3C*, *Figure 3—figure supplement 1C*). In the absence of HH, the apical and lateral regions are slightly more affected than the basal regions (with a 47% and 44% decrease for the former compared to 28% for the latter) (*Figure 3C*, *Figure 3—figure supplement 1C'*). This leads to an increase in the relative abundance of Surf labeled SNAP-SMO in the basal region (associated with a slight but non-significant decrease of the apical fraction) (*Figure 3D*). By contrast, in the presence of HH (P region), the reduction is much more pronounced in the apical region (38%) than in the intermediate (28%) and basal region (which is barely affected, 4%) (*Figure 3C*, *Figure 3—figure supplement 1C'*). This leads to a relative

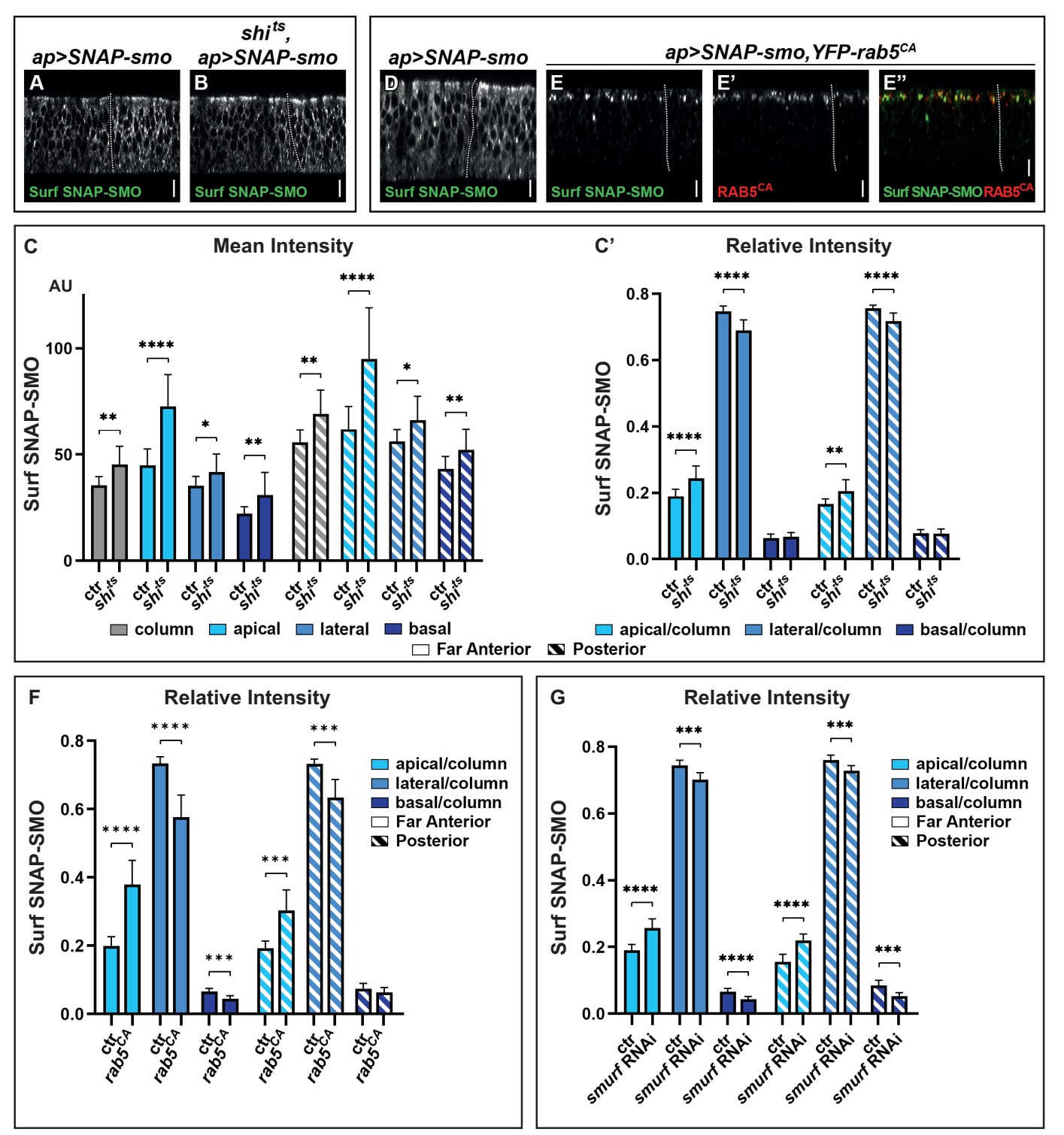

**Figure 2.** Blocking Smoothened (SMO) endocytosis favors its accumulation in the apical region independently of hedgehog (HH). (**A–B**) XZ confocal images of wing imaginal discs (WID) from *shi+; apGal4; UAS SNAP-smo* (A, called *ctr* for control in C and C') or *shi^{ts}; apGal4; UAS SNAP-smo* (B, called *shi^{ts}* in C and C') male flies put at the restrictive temperature for 30 min before dissection, labeling of Surf SNAP-SMO and imaging under the same conditions. Note that here and in all the following figures the images were taken using an SP5 AOBS confocal, 40× oil. (**C–C'**) Quantification of the mean intensities (**C**) and relative intensities (**C'**) of Surf SNAP-SMO in the different regions of the disc (far anterior [FA] and posterior [P]) and along the apico-basal (Ap-Ba) axis, as indicated. The mean intensity is also shown for the whole epithelial column (gray). N=17 and 10 for *ctr* and *shi^{ts}* flies, respectively. (**D–E"**) XZ confocal images of WIDs from *apGal4, Gal80^{ts}; UAS SNAP-smo* (**D**) (ctr) or *apGal4, Gal80^{ts}/+; UAS SNAP-smo/UAS YFP-rab5^{CA}* (**E–E"**) flies

*Figure 2 continued on next page*

*Figure 2 continued*

put at the restrictive temperature for 24 hr (to allow *YFP-rab5$^{CA}$* expression) before dissection and labeling for Surf SNAP-SMO. In the latter case, E″ is a merged image with Surf SNAP-SMO in green and YFP-RAB5$^{CA}$ in red. Note that for better visualization of SMO, imaging of *YFP-rab5$^{CA}$* and ctr discs were not acquired under the same conditions. (**F**) The relative intensity of Surf SNAP-SMO along the Ap-Ba axis in *apGal4, Gal80$^{ts}$; UAS SNAP-smo* ctr discs (called ctr, n=11) and *apGal4, Gal80$^{ts}$/+; UAS SNAP-smo/UAS YFP-rab5$^{CA}$* (called *rab5$^{CA}$*, n=8) discs. (**G**) The relative intensity of Surf SNAP-SMO along the Ap-Ba axis in *apGal4; UAS SNAP-smo* (called ctr, n=20) and *apGal4/UAS smurf RNAi; UAS SNAP-smo* (called *smurf* RNAi, n=6) discs. Note that *YFP-rab5$^{CA}$* or *smurf* RNAi overexpression has a much stronger effect than *shi$^{ts}$* inactivation, reflecting that endocytosis is blocked for a much shorter time in the latter case. Here and in all the following figures, the XZ images correspond to a unique single XZ section.

The online version of this article includes the following source data and figure supplement(s) for figure 2:

**Source data 1.** Mean intensity of Surf SNAP-SMO for control and *shi$^{ts}$* flies.

**Source data 2.** Relative intensity of Surf SNAP-SMO for control and *shi$^{ts}$* flies.

**Source data 3.** Relative intensity of Surf SNAP-SMO for control and RAB5$^{CA}$ flies.

**Source data 4.** Relative intensity of Surf SNAP-SMO for control and *smurf* RNAi flies.

**Figure supplement 1.** Effect of *shi$^{ts}$* and *rab5$^{CA}$* on Surf SNAP-Smoothened (SNAP-SMO).

**Figure supplement 1—source data 1.** Relative intensity of Surf SNAP-SMO in *shibire$^{ts}$* flies at 18°C or 30°C.

**Figure supplement 1—source data 2.** Relative intensity of Surf SNAP-SMO in control and *shibire$^{ts}$* flies at 18°C.

redistribution in the P region (compared to the FA region) of endocytosed Surf labeled SNAP-SMO from the apical to the basal and intermediate regions (*Figure 3D*).

In summary, our data show that, both with and without HH, SMO endocytosis leads to a change in its distribution along the Ap-Ba axis, in favor of the basal and intermediate regions, suggesting that SMO could undergo transcytosis. Moreover, HH reduces the degradation of endocytosed SMO and favors its presence in the intermediate and basal regions. As HH does not dramatically affect SMO endocytosis, this indicates that HH controls the fate of endocytosed SMO by reducing its degradation and increasing its recycling.

## Phosphorylation by the PKA/CKI and FU kinases regulates the Ap-Ba localization of SMO at the cell surface

Given that the PKA and FU kinases positively regulate SMO activation and accumulation at the membrane, we analyzed their effect on the Ap-Ba localization of SMO by looking at forms of SMO mimicking (S to D replacements) or blocking (S to A substitution) these phosphorylations.

First, we looked at the constitutively active SNAP-SMO$^{PKA-SD}$, which mimics SMO fully phosphorylated by PKA and CKI kinases (*Jia et al., 2004*). As expected from previous data with SMO$^{PKA-SD}$ in cultured cells (*Jia et al., 2004*; *Sanial et al., 2017*), Surf SNAP-SMO$^{PKA-SD}$ accumulates at the cell membrane both in the anterior and posterior compartments of the wing disc (*Figure 4A and B*, *Figure 4—figure supplement 1A-A′, B-B′*). Moreover, even in the absence of HH (FA cells), its accumulation in the apical region decreases in favor of the lateral and even more of the basal region, similar to what is seen with Surf SNAP-SMO$^{WT}$ in presence of HH (*Figure 4E–E″*). These effects in the FA region are partially suppressed by mutations that prevent the phosphorylation by the FU kinase, with especially a strong reduction of the basal localization of Surf SNAP-SMO$^{PKA-SD\ FU-SA}$, compared with Surf SNAP-SMO$^{PKA-SD}$ (*Figure 4C and E–E″*, *Figure 4—figure supplement 1C-C′*). In contrast, Surf SNAP-SMO$^{PKA-SD\ FU-SD}$, which accumulates at high levels at the cell surface in both compartments of the discs, shows a further decrease in its accumulation at the apical and lateral cell surface in A cells compared to SMO$^{PKA-SD}$ (*Figure 4D and E–E″*, *Figure 4—figure supplement 1D-D′*). This effect is associated with a strong basal enrichment.

The phosphorylation of SMO by the PKA is known to promote its activation and accordingly, the expression of SNAP-SMO$^{PKA-SD}$ leads to both an anterior expansion of anterior *en* and the CI-A regions, two outcomes of high HH signaling (compare the dorsal region in which SNAP-SMO$^{PKA-SD}$ is expressed to the ctr ventral region in *Figure 4—figure supplement 1F-F″*). In contrast, expression of SMO$^{PKA-SD\ FU-SA}$ under the same conditions led to a reduction in both the anterior *en* and CI-A domains (*Figure 4—figure supplement 1G-G″*).

Overall, these results provide evidence that the phosphorylation by the PKA/CKI recapitulates the effects of HH on Surf SMO Ap-Ba distribution, leading to a relative apical depletion and basolateral enrichment. They also show that this effect is dependent upon further phosphorylation of SMO by

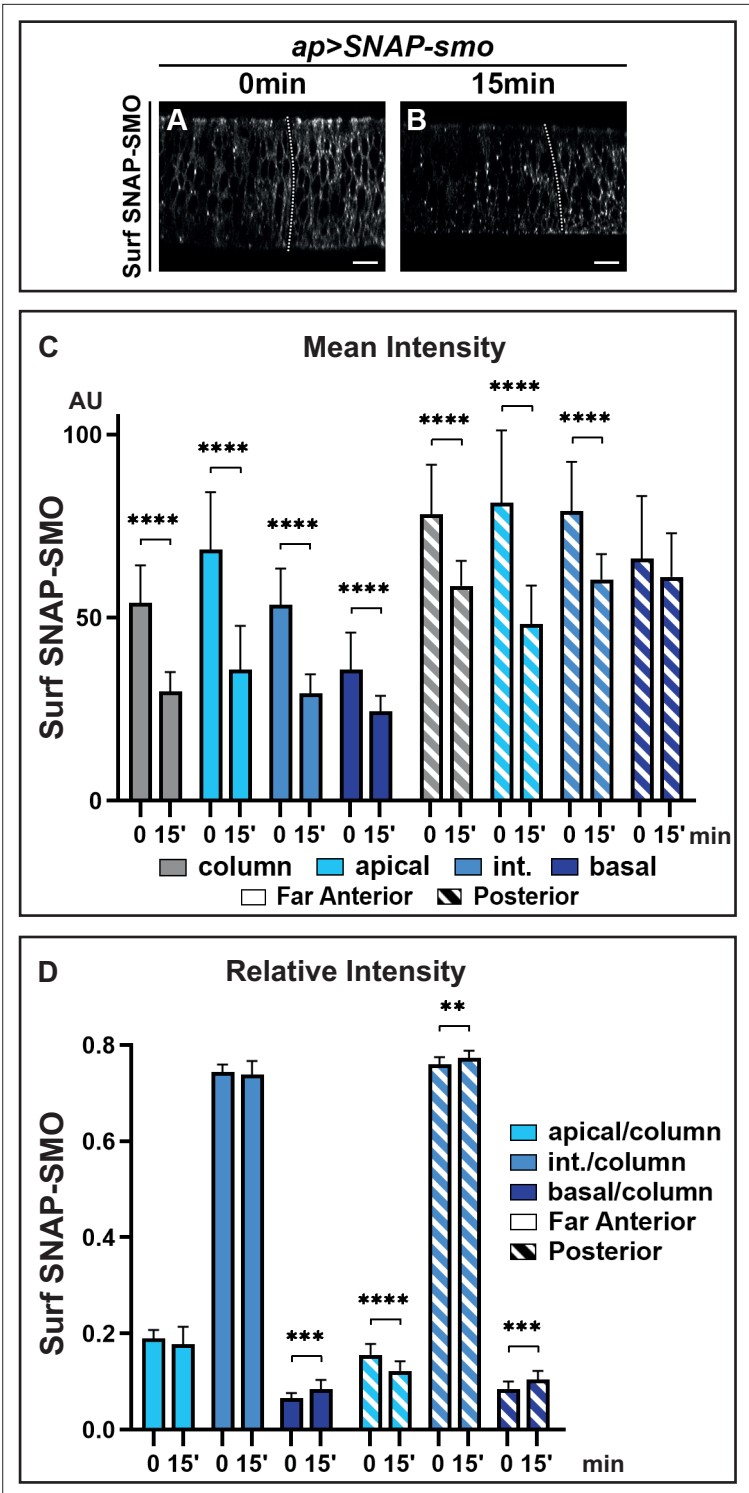

**Figure 3.** Hedgehod (HH) controls the fate of endocytosed Smoothened (SMO), favoring its recycling and leading to its basolateral accumulation. XZ confocal images of dissected wing imaginal discs expressing *UAS SNAP-smo* fixed immediately (0 min) (**A**) or 15 min (**B**) after labeling of Surf SNAP-SMO. Quantification of the mean intensity (**C**) and of the relative intensity (**D**) in the far anterior and posterior regions of the disc and along the apico-basal axis. N=20 for 0 min and 23 for 15 min. Int: intermediate. The discs were treated in the same conditions, all the images were acquired under the same conditions and the dynamic range was normalized to allow a better comparison of Surf SNAP-SMO.

*Figure 3 continued on next page*

*Figure 3 continued*

The online version of this article includes the following source data and figure supplement(s) for figure 3:

**Source data 1.** Mean intensity of Surf SNAP-SMO at T0 and at T15'.

**Source data 2.** Relative intensity of Surf SNAP-SMO at T0 and T15'.

**Figure supplement 1.** The fate of endocytosed SNAP-Smoothened (SNAP-SMO).

**Figure supplement 1—source data 1.** Percentage of decrease in Surf SNAP-SMO between T0 and T15.

FU, which both especially enhances its distribution in the basal region and is required for high HH signaling.

## FU is required first apically and then basally to promote high SMO activity

Our data strongly link the basal localization of SMO and its activation by the FU kinase. FU is present everywhere along the Ap-Ba axis, with some enrichment in the apical region. It is both diffused in the cytoplasm and present in vesicular puncta, some of which colocalize with SMO (*Claret et al., 2007*; *Figure 5—figure supplement 1A, A', B-B"*). To understand where FU acts on SMO, we trapped it in the apical or in the basolateral region of the cells using the GRAB bipartite system (*Harmansa et al., 2017*). In this method, GFP tagged FU (GFP-FU, known to be fully functional *Malpel et al., 2007*) and whose Ap-Ba distribution is similar to that of endogenous FU (*Figure 5—figure supplement 1C*), is trapped in either the apical or basolateral domain via its binding to an intracellular GFP nanobody fused to an apical (T48) or basolateral (NVR1) transmembrane domain. For that purpose, we expressed both GFP-FU and its trap in the dorsal part of the disc and we immunolabeled endogenous SMO (*Figure 5*, note that in B–B"' and D–D"', the Z sections correspond to anterior YZ sections across the dorso-ventral axis, with the ventral compartment serving as an internal control).

As expected, expression of *GFP-fu* with *mCherry (mche)-T48* leads to strong apical relocalization of GFP-FU (compare *Figure 5B'* to *Figure 5—figure supplement 1C*), while its coexpression with mche-NVR1 leads to its basolateral enrichment (compare *Figure 5D'* to *Figure 5—figure supplement 1C*). Apical tethering of GFP-FU (with mche-T48) promotes the accumulation of SMO both in the anterior (*Figure 5A" and B"*, *Figure 5—figure supplement 2D*) and posterior (*Figure 5A"'*, *Figure 5—figure supplement 2A"', D'*) compartments. This effect is homogeneous along the Ap-Ba axis, with only a slight increase in the basal relative distribution of SMO (*Figure 5—figure supplement 2D"*). Importantly, a similar effect is seen with Surf SNAP-SMO (expressed at endogenous levels from the BAC construct), indicating that T48-tethered GFP-FU stabilizes SMO at the cell surface (*Figure 5—figure supplement 2C-C"'*). By contrast, basolateral trapping of GFP-FU (with mChe-NVR1) has only a slight effect on SMO accumulation (*Figure 5C"–D"* and *Figure 5—figure supplement 2B"'*) and seems to induce its vesicular localization.

Next, we tested the effects of trapping GFP-FU with mche-T48 or NVR1 on HH signaling. We monitored the accumulation of CI and the expression of the low-HH target *dpp* (using a *dpp-LacZ* reporter, *dpp-Z*), the medium/high-HH target *ptc*, and the high-HH target, *en*. While expression of GFP-FU alone has no effect (*Claret et al., 2007*), coexpression of GFP-FU with mche-T48 leads to the ectopic activation of medium levels of HH signaling as it promotes the ectopic expression of *dpp-Z* (*Figure 6—figure supplement 1A*) and *ptc* throughout the whole anterior compartment (although its expression at the A/P border is slightly reduced) (*Figure 6A*). On the other hand, high-level HH signaling was reduced as *en* expression was decreased near the A/P border but expanded at low levels throughout the anterior region (*Figure 6B*). By contrast, trapping GFP-FU with mche-NVR1 has no effect on *dppZ* and *ptc* expression and only weakly reduces the anterior *en* expression (*Figure 6C and D*, *Figure 6—figure supplement 1B*).

To ensure that these effects were not due to an indirect effect of mche-T48-trapped GFP-FU on endogenous FU, we repeated this experiment in a *fu* mutant background. As all currently used *fu* mutants display complex genetic interactions suggesting that they may not be null alleles, we knocked out the *fu* gene by CRISPR, leading to a deletion (called *fu^{KO}*) that removed the entire *fu* transcribed region, except for the 3'UTR region (*Figure 6—figure supplement 1C*). This mutation leads to the total suppression of anterior *en* expression, and a very strong reduction of *ptc* expression (see the ventral region of the discs shown in *Figure 6E, F, G and H*). These effects are similar but stronger

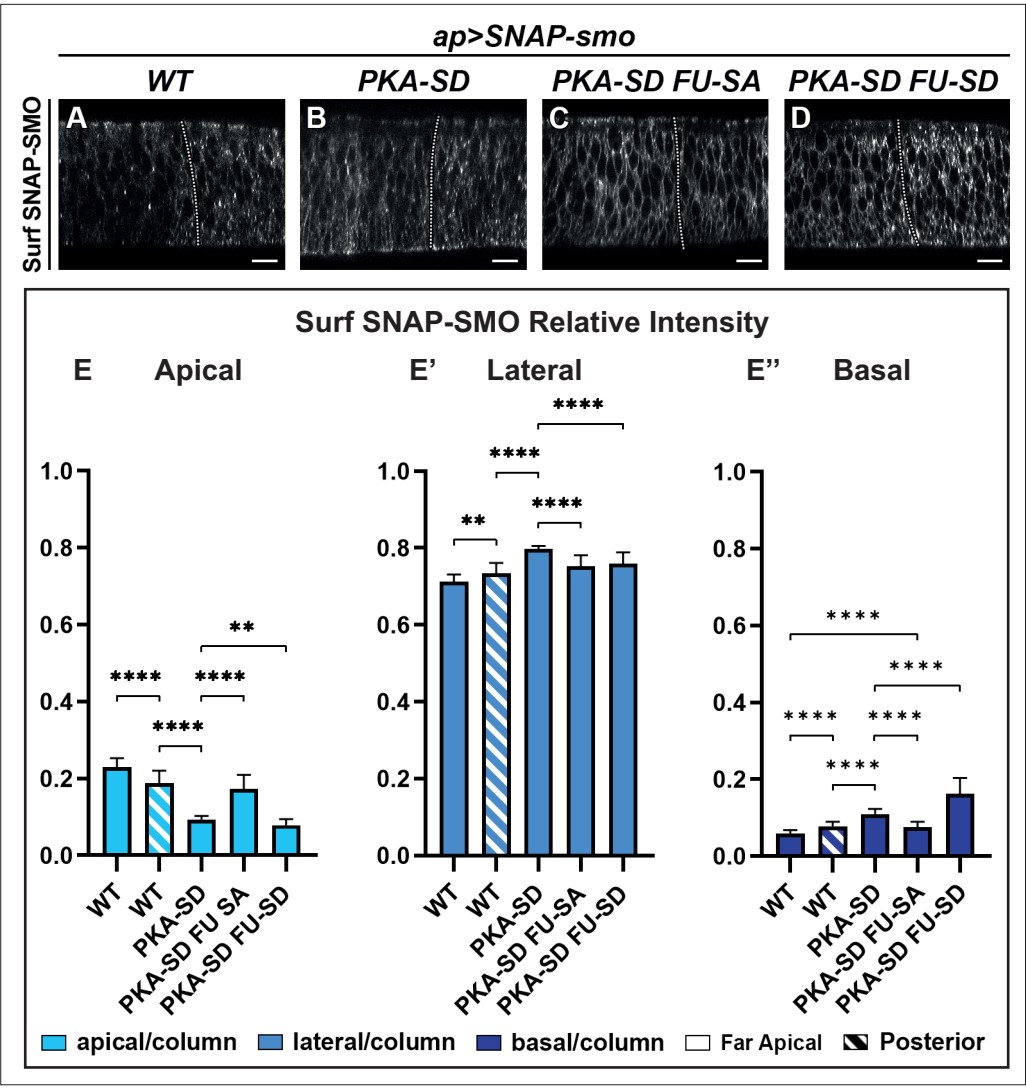

**Figure 4.** The apico-basal (Ap-Ba) distribution of Smoothened (SMO) is controlled by phosphorylation by the protein kinase A (PKA)/ casein kinase I (CKI) and Fused (FU) kinases. (**A–D**) Confocal images of wing imaginal discs labeled for Surf SNAP-SMO[WT] (**A**), SNAP-SMO[PKA-SD] (**B**), SNAP-SMO[PKA-SD FU-SA] (**C**), and SNAP-SMO[PKA-SD FU-SD] (**D**). Note that for better visualization of SMO, imaging of the different forms of SNAP-SMO was not done under the same conditions. See also the corresponding false-color images in *Figure 4—figure supplement 1A*. (**E–E"**) Quantification of the Ap-Ba distribution of the different forms of SNAP-SMO, as indicated. N=21 WT, 10 PKA-SD, 17 PKA-SD FU-SA, and 18 PKA-SD FU-SD discs, respectively. (See *Figure 4—figure supplement 1E*) showing the reproducibility of Surf SNAP-SMO[WT] distribution. Note that the effects of the *PKA-SD* mutations are even stronger than the effect of hedgehog (HH) on SNAP-SMO[WT], probably because in presence of HH, only a fraction of the SMO population is phosphorylated (*Sanial et al., 2017*).

The online version of this article includes the following source data and figure supplement(s) for figure 4:

**Source data 1.** Relative intensity of Surf SNAP-SMO for different SMO mutants.

**Figure supplement 1.** Effect of the phosphorylation of Smoothened (SMO) by the protein kinase A (PKA) and Fused (FU) kinases on its accumulation and signaling activity.

**Figure supplement 1—source data 1.** Relative intensity of Surf SNAP-SMO in WT1 and WT2.

than what was shown for strong *fu* mutants (*Alves et al., 1998*). In this context, expression of GFP-FU trapped with mche-T48 has a similar effect to that in presence of the endogenous functional FU protein: ectopic anterior *ptc* expression, reduced *en* expression near the A/P, (see the dorsal region of the discs shown in *Figure 6E and F*). Strikingly, while the expression of GFP-FU alone suppresses the effect of the *fu^KO* mutation on HH signaling with the restoration of *ptc* and anterior *en* expression

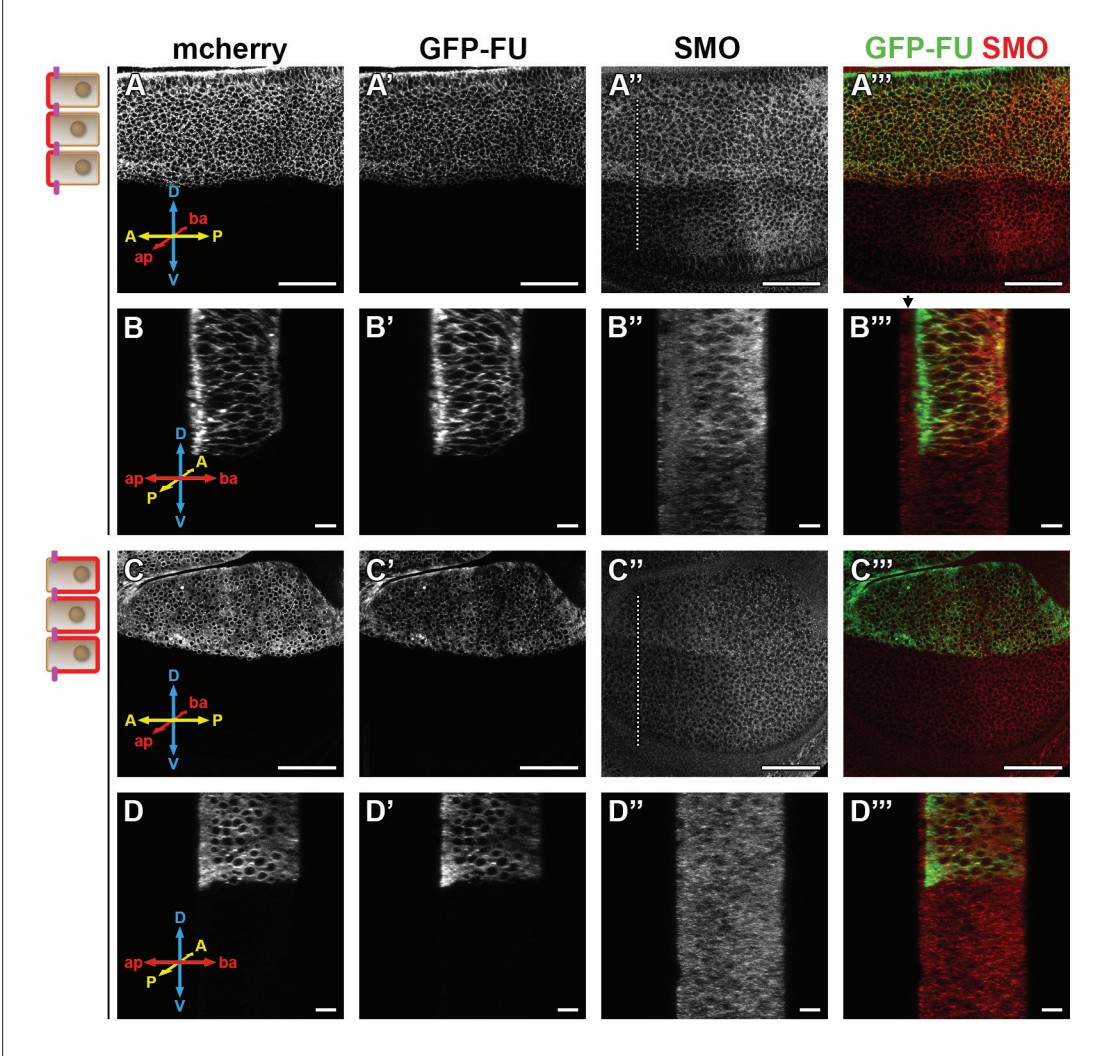

**Figure 5.** Trapping the Fused (FU) kinase to the apical region promotes the stabilization of Smoothened (SMO) at the cell surface. Confocal images of wing imaginal discs coexpressing (using the apGal4 driver) *GFP-fu* and *T48* or *NVR1* fused to the *mcherry* (*mche*). XY sections are shown in (**A–A"' and C–C"'**), anterior YZ sections in (**B–B"' and D–D"'**). The mche is shown in (**A, B, C, and D**), GFP-FU in A', B' C', and D', green in the merged images A"', **B"', C"', and D"'** and immunolabeled endogenous SMO in A", B", C" and D", red in the merged images A"', **B"', C"', and D"'**. Note that contrary to SNAP-SMO, endogenous SMO is also present in the peripodial membrane (black arrow above the SMO images). **D**: dorsal, V: ventral, A: anterior, P: posterior, ap: apical, and ba: basal. Here and in **Figure 6**, the scale bar for XY section represents 50 µm.

The online version of this article includes the following source data and figure supplement(s) for figure 5:

**Figure supplement 1.** Fused (FU) and Smoothened (SMO) colocalization.

**Figure supplement 2.** Effect on the subcellular localization of Smoothened (SMO) of trapping the Fused (FU) kinase to the apical (ap) or basolateral region.

**Figure supplement 2—source data 1.** Relative and mean intensity of endogenous SMO in T48 GFP-FU flies.

(see the dorsal region of the discs shown in **Figure 6G and H**), it was unable to promote the effect of T48-tethered GFP-FU on *ptc* and anterior *en*. These data indicate that the effects seen when GFP-fu is coexpressed with mche-T48 are indeed due to its apical tethering and do not require the presence of endogenous FU.

In summary, anchoring FU to the apical membrane, but not to the basolateral one, increases the levels of SMO at the cell surface. It also leads to its constitutive activity, promoting low-medium HH signaling but blocking very-high HH signaling. This suggests (i) that only apical FU can activate SMO and stabilize it at the cell surface and (ii) that both the high HH-induced basal enrichment of SMO and the expression of high HH targets may require a second input of FU on SMO in the basolateral region.

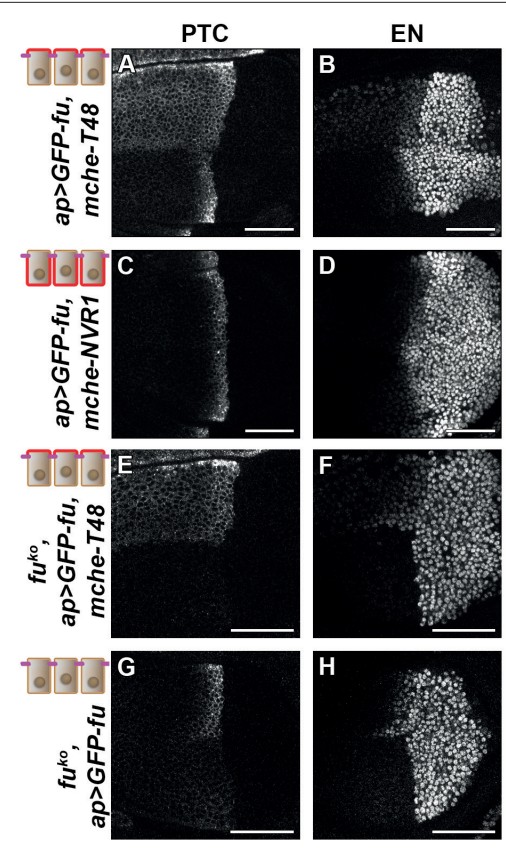

**Figure 6.** Trapping the Fused (FU) kinase to the apical region is sufficient to promote medium but not high hedgehog (HH)/ Smoothened (SMO) signaling. Confocal XY images of discs expressing GFP-fused with mche-T48 in $fu^{WT}$ (**A–B**) or knockout ($fu^{KO}$) context (**E–F**), with mche-NVR1 in $fu^{WT}$ (**C–D**) or alone in $fu^{KO}$ context (**G–H**) and immunolabeled for Patched (PTC) (**A, C, E, and G**) or EN (**B, D, F, and H**). In (**A–D**), the normal pattern of expression of these genes is visible in the ventral region of the discs.

The online version of this article includes the following figure supplement(s) for figure 6:

**Figure supplement 1.** Effect of trapping the Fused (FU) kinase on *decapentaplegic (dpp)* expression.

## Discussion

Understanding how HH controls the fate of SMO in a polarized epithelium is central to understanding how this GPCR can be activated both in physiological and pathological conditions. Here we provide evidence that supports a model (*Figure 7*) whereby (i) SMO is initially addressed to the apical membrane before being transcytosed to the basolateral region, (ii) HH controls the post-endocytic fate of SMO likely by enhancing its recycling, especially in the basolateral region, (iii) very high levels of HH favors local trapping of SMO in the most basal region, and finally that (iv) these effects rely on an SMO phosphorylation-barcode determined by the sequential action of the PKA/CKI and FU kinases, with FU acting in a two-step manner.

The stabilization of SMO induced by HH could result from a reduction of its internalization or of its degradation after internalization. Our results indicate SMO endocytosis is little affected by HH and that HH acts on endocytosed SMO, shifting its fate toward recycling rather than degradation. This involves the phosphorylation of SMO by the PKA/CKI, whose effects are further enhanced by a secondary action of FU. Notably, phosphorylation of the β-adrenergic receptor by the PKA has also been shown to increase its recycling to promote its resensitization (*Gardner et al., 2004*).

We have previously provided evidence for a double positive SMO-FU feedback loop behind 'high HH' signaling: FU recruitment at the plasma membrane by SMO leads to the first level of FU activation, which in turn further activates SMO, which further increases FU activation. Here, tethering of FU to the apical membrane—but not to the basolateral one—is sufficient to ectopically promote both the stabilization of SMO at the cell membrane and the activation of low/medium HH targets. However, in contrast to what is seen in presence of very high levels of HH or when SMO is fully hyperphosphorylated (SMO$^{PKA-SD\ FU-SD}$), apical FU does not promote high levels of HH signaling and does not lead to a basal enrichment of SMO. Together, these data support the existence of the second effect of FU on SMO, which would occur in the basolateral region and lead to a basal accumulation of SMO, promoting very high HH signaling. Note that the aPKC was also reported to positively modulate SMO activity and to favor (directly or indirectly) its basolateral localization (*Jiang et al., 2014*). However, contrarily to the phosphorylation of SMO by FU, the phosphorylation by the aPKC does not seem to affect the 'high HH'-dependent basal localization of SMO, nor 'high HH' signaling.

Although the entire basolateral membrane is overall considered as a unique membrane domain in which proteins and lipids freely diffuse, many examples of membrane subregionalization exist (for review see *Trimble and Grinstein, 2015*). Here, we provide evidence that hyperactivated SMO can be enriched in the most basal region. It could in part be due to a partial reduction of its basal endocytosis (as suggested by our experiments), but it likely also involves other mechanisms that were shown to restrain the localization of transmembrane proteins (for review see *Trimble and Grinstein, 2015*). For

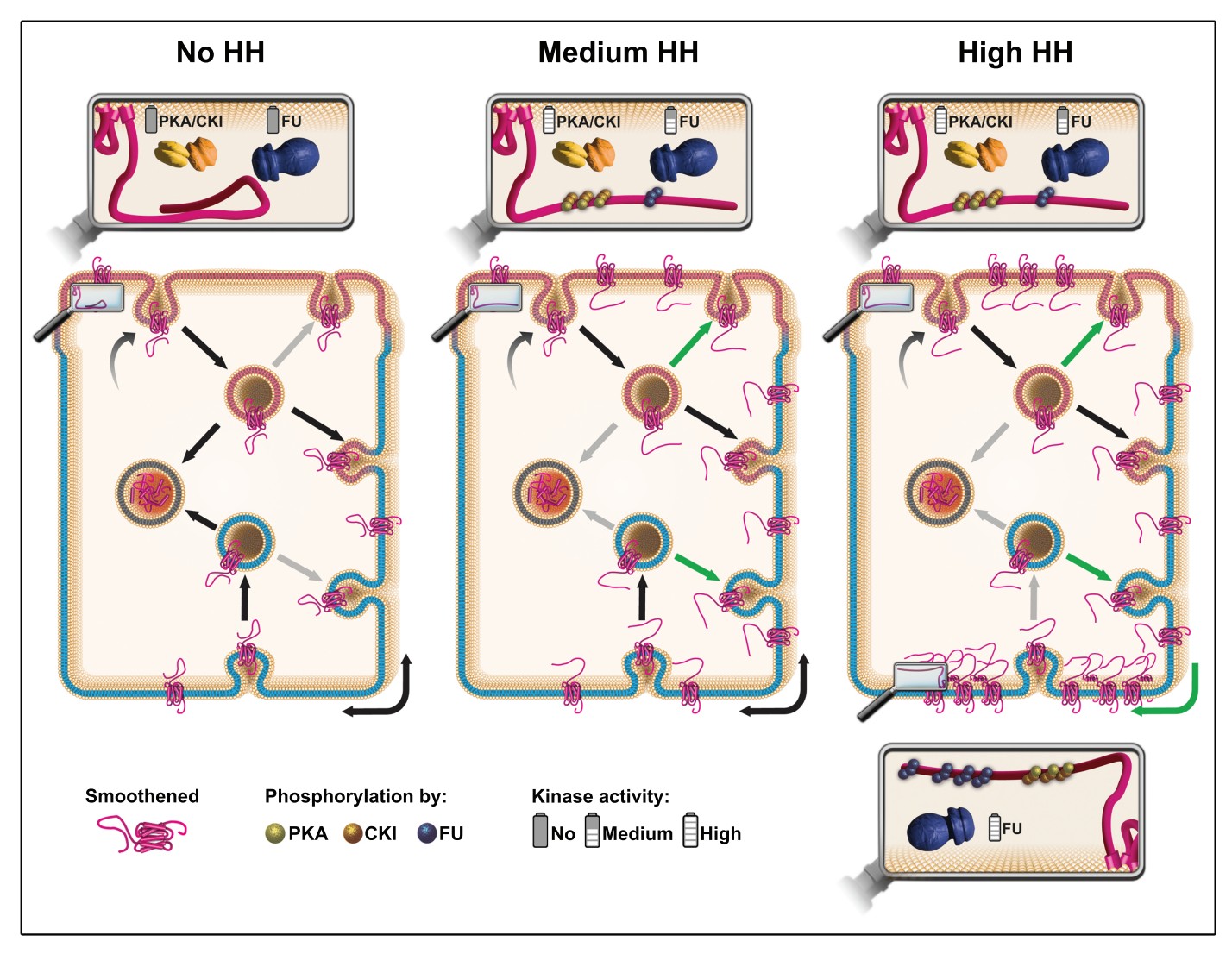

**Figure 7.** Model: hedgehog (HH) controls the fate of endocytosed Smoothened (SMO) via a multikinase phosphorylation barcode. SMO is initially targeted (curved arrow) to the apical plasma membrane (presented in pink), where it is endocytosed (vesicle on the top, with a pink membrane). Endocytosed SMO can then be sent to the lysosome for degradation (vesicle on the left with gray membrane), recycled to the apical membrane, or transcytosed to the basolateral membrane (represented in blue), where it can, in turn, be re-endocytosed (followed by recycling or degradation). HH controls several of these steps via the graded action of the proteing kinase A (PKA)/casein kinase I (CKI) and Fused (FU) kinases. The double arrow represents the diffusion of SMO in the basolateral plasma membrane. In absence of HH (left panel), a large fraction of endocytosed SMO is targeted for degradation at the expense of its recycling (reduced recycling is indicated by gray arrows). In that situation, the C-terminal cytoplasmic domain of SMO is not phosphorylated and adopts a closed, inactive conformation. Conversely, in presence of HH (see middle and right panels) the degradation of endocytosed SMO is reduced (gray arrows), in favor of its recycling (green arrows), especially in the basolateral region. These effects of HH lead to an increased accumulation of SMO at the cell surface and require the phosphorylation of the C-terminal cytoplasmic domain of SMO by the PKA/CKI. This triggers a change in the conformation of this region, which is associated with the activation of SMO. In the presence of intermediate levels of HH (middle panel), SMO accumulation and/or activation reaches a threshold that in turn activates FU, which then retroacts on SMO, further enhancing the stabilizing and activating effects of the PKA/CKI kinases. The fact that FU is required for that effect in the apical region, suggests that the PKA/CKI also acts in this domain. In the presence of high levels of HH (right panel), FU is further activated and promotes the enrichment of SMO in the most basal region by phosphorylating its C-terminal cytoplasmic domain, which likely increases its clustering. This favors SMO's trapping in the most basal region, which leads to high levels of HH signaling. The green arrow on the bottom right corner indicates a putative diffusion trapping mechanism, the gray bottom arrow indicates that recycling may be subsequently reduced.

instance, it could involve an active oriented displacement of endocytic vesicles carrying SMO and FU directly to the basal domain by the kinesin COS2, which is known to be required for high HH signaling and to transport SMO and FU along microtubules in cultured fly cells (*Farzan et al., 2008*). Alternatively, diffusion trapping or partitioning phenomena (for review see *Trimble and Grinstein, 2015*) are also known to lead to local protein enrichment, for instance in axons (*Ashby et al., 2006*). Here, the changes in SMO electrostatic charges, conformation, and/or clustering, which result from its hyperphosphorylation, could favor such processes (*Shi et al., 2013*; *Zhao et al., 2007*).

Regardless of the mechanism leading to this basal subpopulation of SMO, our data show that its presence is correlated to high levels of SMO activation and the basal gradient of HH. Although we cannot exclude that this basal localization is a consequence rather than a cause of SMO activation, for instance, a desensitization mechanism, our data along with published results strongly suggest that it is crucial to promote high HH signaling. Indeed, our results strongly connect SMO basal localization to its 'high activation' as: (i) high HH leads to SMO basal localization, (ii) the phosphorylation of SMO by FU is required for both SMO basal accumulation and its highest level of signaling activity, and (iii) on the contrary blocking FU in the apical region reduces both events. We propose that in presence of high levels of HH, SMO could be trapped in basal specialized lipid microdomains that enhance signaling, similarly to what has been shown for the regulation of several GPCR by lipids rafts (for review see *Villar et al., 2016*). This possibility is supported by many reports showing that both in flies and in mammals SMO responds to changes in its lipid environment (for review see *Radhakrishnan et al., 2020* and *Zhang et al., 2021*) with an emerging key role of accessible cholesterol (*Kinnebrew et al., 2021*). Notably, in *Drosophila* WID cells, SMO also was shown to relocalize in response to HH to cholesterol-rich raft lipid microdomains in the plasma membrane, where it forms higher-order structures (e.g. oligomers) that are required for high HH signaling, but the Ap-Ba localization of these rafts was not addressed (*Shi et al., 2013*). Such microdomains could constitute signaling platforms acting on SMO structure (*Zhao et al., 2021*) enhancing the oligomerization of SMO and the HTC (*Shi et al., 2011*) and /or the interaction of SMO with specific HH signaling protein(s) (as seen with the EvC proteins in the cilia, see below [*Dorn et al., 2012*; *Yang et al., 2012*]).

Several lines of evidence point toward the conservation of a diffusion-trapping mechanism that would lead to the activation of SMO via its subcompartmentalization in specific membrane domains of the plasma membrane. Indeed, in response to Sonic HH, the entry of SMO in the primary cilium of mammalian cells, which also depends on its phosphorylation (*Chen and Jiang, 2013*), was shown to involve the lateral diffusion of SMO from the cell body into the cilium membrane (*Milenkovic et al., 2009*) and is followed by its spatial restriction in a ciliary distinct compartment named the EvC zone (*Dorn et al., 2012*; *Yang et al., 2012*). This involves the phosphorylation-dependent interaction of SMO with two components of this zone, EvC and EvC2, both acting downstream of SMO to alleviate the negative effects exerted by the Suppressor of FU (SUFU). In that respect, it is worth noting, that in the *Drosophila* WID, the negative effects of SUFU need to be suppressed by FU for high 'HH signaling' (*Alves et al., 1998*).

## Materials and methods

### *Drosophila* strains and genetics

All *smo* transgenes, except for the *BAC (CH322-98K24) SNAP-smo,* which was introduced at the landing site 9725 on 3 R (at 75A10), were introduced into the landing site (9738) on 3 R (at 99F8), using the PhiC31 integration system to ensure that they are expressed at similar levels (*Bateman et al., 2006*) by BestGene Inc the *fu*[KO] mutant was generated by inDroso functional genomics using CRISPR/Cas9 mediated genome editing (*Jinek et al., 2012*).

The flies were kept at 25°C with three exceptions: for *shibire*[ts] experiments, the flies were kept at 18°C before being switched, at the third instar larval stage, to 30°C for 30 min; for *rab5*[CA] and *hh*[ts2] experiments, flies were kept at 18°C for 7 to 8 days before being switched to 29°C for 24 hr.

For the genotypes of the *Drosophila* strains used here, see Appendix 1.

### WID SNAP labeling

For surface labeling: third instar larvae were dissected in Shields and Sang M3 Insect cell complete medium and incubated for 10 min at 25°C (or at 29°C for *BAC (CH322-98K24) SNAP-smo* experiments

and 30°C for *shi^ts* experiments) with the SNAP-Surface Alexa Fluor (NEB) in complete medium, then fixed 20 min at room temperature in 4% paraformaldehyde, and washed three times 10 min in PBS + 0.3% Triton (PBST). This was followed by immunolabeling.

For intracellular labeling: after surface labeling and fixation, discs were incubated with SNAP-Cell TMR-Star (NEB) and SNAP-Surface Block (NEB) for 20 min at 25°C (for intracellular labeling), followed by immunolabeling.

For immunolabeling, discs were incubated with the primary antibody overnight at 4°C. They were then washed three times with PBST and incubated for 2 hr at RT with the secondary antibody in PBST before another three washes of 10 min in PBST. Discs were then mounted in Citifluor (Biovalley).

For the different concentrations and details on the chemicals used see Appendix 1.

## Fly wings

For wing harvesting, flies were collected in ethanol 70%, the wings were then dissected in water, and mounted in Hoyer's medium. Pictures were taken with a Zeiss Lumar stereomicroscope and the AxioVision software.

## Analysis of SMO trafficking

For the pulse-chase experiments, dissected WIDs were incubated for 10 min at 25°C with SNAP-Surface Alexa Fluor, rinsed, and incubated for 15 min with SNAP-Surface Block before fixation and immunolabeling.

## Image acquisition, processing, and quantification

Images in *Figure 1* and *Figure 5—figure supplement 1B-B"* were acquired using the confocal Zeiss LSM980 spectral Airyscan 2, 63× oil. All the other images were acquired using the confocal Leica confocal SP5 AOBS, 40× oil. XZ and YZ stacks of discs were acquired with sections every 1 μm. Microscope settings were chosen to allow the highest fluorescence levels to be imaged under non-saturating conditions. Image data were processed and quantified using ImageJ software (National Institute of Health).

An ImageJ macro was designed to quantify SNAP-SMO fluorescence from the Z projections (average intensity) of the stack (8 sections). Firstly, the macro determines the shape and limits of the disc. It then asks the user to select a point in the disc (selected at the A/P boundary) from which rectangular regions from apical to basal, with the same width, will be drawn across the disc. Following this, each region is divided into three smaller ones: the apical/subapical, lateral, and basal subregions. The macro considers apical/subapical and basal subregions to be 15% and 10%, respectively, of the disc's thickness. The 15% value was fixed based on the immunostaining of the septate junctions by DLG. The 10% value is arbitrary and based on image analysis. The macro proceeds to measure the raw integrated density and the mean density of the different regions. Finally, with the help of CI immunostaining, four regions are selected across the disc (as shown in *Figure 1—figure supplement 2C*) to do the quantification. For macro description, see Appendix 2.

## Statistics and data representation

Statistical analysis was carried out using GraphPad Prism 9. The sample size was chosen large enough ($n \geq 8$) to allow assessment of the statistical significance. Sample numbers are indicated in the figure and source data for each experiment. N-numbers indicate biological replicates, meaning the number of biological specimens evaluated (e.g. the number of wing discs). When comparing the A and P compartments within the same disc a paired t-test (for the mean intensities) or a Wilcoxon matched-pairs signed rank test (for the relative intensities) was used (*Figure 1*). When comparing different discs (in all other figures), a Mann-Whitney test was used. The p-values are shown in the corresponding source data.

## Plasmids

All expression vectors, except for the *BAC (CH322-98K24) SNAP-smo* and the *fu^KO* mutant, were constructed by the Gateway recombination method (Invitrogen). The *BAC (CH322-98K24) SNAP-smo* was generated from an attB-P[acman]-Ap BAC using recombineering mediated gap-repair (*Venken et al., 2009*). All mutated regions were verified by sequencing.

We used the Gateway Technology (Invitrogen following the manufacturer's instructions) to introduce the *SNAP-smo*PKA-SD, *SNAP-smo*PKA-SD FU-SA, or *SNAP-smo*PKA PKA-SD FU-SD transgenes in the vector *pUASt-GW-attB* (constructed by A Brigui by insertion of the GW recombination cassette C3 at the EcoRI site of the pUASt-attB plasmid [GI EF362409]) for PhiC31 germline transformation, respectively. Prior to that, the PCR products obtained from the coding sequence (without the termination codon) of a *smo* wild type cDNA were inserted into *pENTR/D-TOPO* by directional TOPO Cloning. Mutations leading to the S to A and S to D changes of the PKA/CKI sites were inserted into *pENTR/D-TOPO-snap-smo* by replacement of a region with a similar region coming from *smo*PKA-SD/SA from **Jia et al., 2004**, leading to *pENTR/D-TOPO-snap-smo*PKA-SA/PKA-SD. The mutations leading to the S to A and S to D replacements of the FU phosphosites were introduced into *pENTR/D-TOPO-snap smo*PKA-SD by replacement of a region by a similar region coming from *pENTR/D-TOPO-smo smo*FU-SD/SA (**Sanial et al., 2017**) leading to *pENTR/D-TOPO-snap-smo*PKA-SD FU-SA and *pENTR/D-TOPO-snap-smo*PKA-SD FU-SD. All constructs were checked by sequencing the fragments produced by PCR and their junctions.

The BAC transgene was generated from an attB-P[acman]-Ap BAC. Briefly, a functional *smo* BAC (CH322-98K24) was modified using recombineering mediated gap-repair (**Venken et al., 2009**) to introduce *snap* cDNA at the N-terminus of *smo* after the codon encoding the Serine 33 at the end of the signal peptide sequence (**Alcedo et al., 1996**). *pENTR/D-TOPO-snap-smo* was used as a template to amplify the *snap* cDNA. For *snap* cDNA insertion into *smo* mRNA (present in CH322-98K24), first, the primers rpsL-neo/smo mRNA 360/F and rpsL-neo/smo mRNA 462/R (see Appendix File 1 key resource table) were used to amplify the *rpsL-neo* cassette. Second, the cassette was replaced by *snap* cDNA amplified using the primers pEnSnapSmo/smo mRNA 360/F and pEnSnapSmo/smo mRNA 462/R (see M&M table). Insertions were confirmed using different sequencing primers: smo mRNA 289/Seq/F, smo mRNA 512/Seq/R, Rpsl/neo/273/Seq/F, and snap/203/Seq/F (see Appendix File 1 key resource table).

For *fu*KO mutant generation, two guide RNAs (gRNA) and a double-strand DNA plasmid donor containing the fluorescent marker DsRed, were used to lead homology-directed repair in the *fu* locus. The gRNA1 anneals 545 bp upstream the ATG, while the gRNA2 anneals 26 bp upstream the TAG. Approximately, 2.6 Kb were removed, including the 5'UTR but not the 3'UTR, and replaced with the coding sequence of DsRed fluorescent marker, which provides a marker to distinguish *fu*KO larvae.

## Acknowledgements

We are grateful to Drs M Crozatier, A Guichet, D Hipfner, J Jian, P Therond, F Schweisguth for generously sharing their fly lines and reagents; to A Benhmerah, A Guichet and S Léon and our colleagues from the Institut Jacques Monod for insightful discussions, Lisa Barbuglio, Andréa Mialet and Severine Nozownik for their technical help. We are very grateful to Drs. G D'Angelo and R Holmgren for sharing their expertise and for their insightful advice. The Apa 1, 4D9, 4F3, and 20C6 monoclonal antibodies were obtained from the Developmental Studies Hybridoma Bank, created by the NICHD of the NIH and maintained at The University of Iowa, Department of Biology, Iowa City, IA 52242. *Drosophila* embryo injections were carried out by BestGene Inc and by inDroso. We acknowledge the ImagoSeine core facility of Institut Jacques Monod, member of France-BioImaging (ANR-10-INBS-04) and certified IBiSA. This work was supported by the Centre National de la Recherche Scientifique CNRS, the Université de Paris, and the Fondation ARC pour la recherche sur le Cancer (JA20191209287). MGA was supported by the Université de Paris (CNRS and the Ecole Universitaire Génétique et Epigénétique Nouvelle Ecole (EUR G.E.N.E)).

# Additional information

## Funding

| Funder | Grant reference number | Author |
|---|---|---|
| Institut des sciences biologiques | | Marina Gonçalves Antunes<br>Matthieu Sanial<br>Vincent Contremoulins<br>Sandra Carvalho<br>Anne Plessis<br>Isabelle Becam |
| Fondation ARC pour la Recherche sur le Cancer | JA20191209287 | Anne Plessis |
| EurGene | | Marina Gonçalves Antunes |
| Universite de Paris Cité | | Marina Gonçalves Antunes |

The funders had no role in study design, data collection and interpretation, or the decision to submit the work for publication.

## Author contributions

Marina Gonçalves Antunes, Conceptualization, Formal analysis, Validation, Investigation, Methodology, Writing - review and editing; Matthieu Sanial, Formal analysis, Investigation, Visualization, Methodology, Writing - review and editing; Vincent Contremoulins, Sandra Carvalho, Investigation; Anne Plessis, Isabelle Becam, Conceptualization, Formal analysis, Supervision, Writing - original draft, Writing - review and editing

## Author ORCIDs

Marina Gonçalves Antunes http://orcid.org/0000-0003-3818-4434
Matthieu Sanial http://orcid.org/0000-0002-0301-4710
Sandra Carvalho http://orcid.org/0000-0002-8654-0539
Anne Plessis http://orcid.org/0000-0001-9193-7031
Isabelle Becam http://orcid.org/0000-0002-4464-7880

## Decision letter and Author response

Decision letter https://doi.org/10.7554/eLife.79843.sa1
Author response https://doi.org/10.7554/eLife.79843.sa2

# Additional files

## Supplementary files
• MDAR checklist
• Source code 1. Fiji Macro for the quantification.

## Data availability

All data generated or analysed during this study are included in the source data. The script is provided in Source Code 1.

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

# Appendix 1

## Appendix 1—key resources table

| Reagent type (species) or resource | Designation | Source or reference | Identifiers | Additional information |
|---|---|---|---|---|
| Genetic reagent (*D. melanogaster*) | apGal4 | **Weihe et al., 2001** | FLYB: FBtp0084979 | FlyBase symbol: P{ap-GAL4.U} |
| Genetic reagent (*D. melanogaster*) | UAS SNAP-smo$^{WT}$ | **Sanial et al., 2017** | | |
| Genetic reagent (*D. melanogaster*) | UAS SNAP-smo$^{PKA-SD}$ | This paper | | Generated by BestGene Inc using the PhiC31 integration system |
| Genetic reagent (*D. melanogaster*) | UAS SNAP-smo$^{PKA-SD}_{FU-SA}$ | This paper | | Generated by BestGene Inc using the PhiC31 integration system |
| Genetic reagent (*D. melanogaster*) | UAS SNAP-smo$^{PKA-SD}_{FU-SD}$ | This paper | | Generated by BestGene Inc using the PhiC31 integration system |
| Genetic reagent (*D. melanogaster*) | hh$^{ts}$ | **Heemskerk and DiNardo, 1994** | FLYB: FBal0031490 | |
| Genetic reagent (*D. melanogaster*) | shibire$^{ts}$ | **van der Bliek and Meyerowitz, 1991** | FLYB: FBal0015610 | |
| Genetic reagent (*D. melanogaster*) | apGal4 Gal80$^{ts}$ | Gift from F Schweisguth | | |
| Genetic reagent (*D. melanogaster*) | UAS-YFP-rab5$^{CA}$ | Hugo J. Bellen, Baylor College of Medicine | RRID:BDSC_9773 | |
| Genetic reagent (*D. melanogaster*) | UAS smurf RNAi | **Perkins et al., 2015** | RRID:BDSC_40905 | |
| Genetic reagent (*D. melanogaster*) | UAS mCherry-NVR1-GFP nanobody | **Harmansa et al., 2017** | RRID:BDSC_68175 | |
| Genetic reagent (*D. melanogaster*) | UAS mCherry-T48-GFP nanobody | **Harmansa et al., 2017** | RRID:BDSC_68178 | |
| Genetic reagent (*D. melanogaster*) | UAS GFP-fused | **Ruel et al., 2003** | | |
| Genetic reagent (*D. melanogaster*) | dpp-LacZ, apGal4 | Gift from R Holmgren | | *dpp-LacZ* RRID:BDSC_12379 |
| Genetic reagent (*D. melanogaster*) | w$^{1118}$ | **Hazelrigg et al., 1984** | FLYB: FBal0018186 | FlyBase symbol: Dmel\w$^{1118}$ |
| Genetic reagent (*D. melanogaster*) | BAC SNAP- smo | This paper | | *snap* insertion via recombineering mediated gap-repair. Fly generated by BestGene Inc using the PhiC31 integration system. |
| Genetic reagent (*D. melanogaster*) | fu$^{KO}$/FM0Bar | This paper | | *fu* knockout (KO) mutant fly generated via CRISPR/Cas9 by inDROSO |
| Antibody | anti-PTC (mouse monoclonal) | DSHB, **Martín et al., 2001** | DSHB Cat# *Drosophila* Ptc (Apa 1), RRID:AB_528441 | IF(1:50) |
| Antibody | anti-EN (mouse monoclonal) | DSHB, University of Iowa, USA, **Patel et al., 1989** | DSHB Cat# 4D9 anti-engrailed/invected, RRID:AB_528224 | IF(1:50) |
| Antibody | anti-DLG (mouse monoclonal) | DSHB, **Parnas et al., 2001** | DSHB Cat# 4F3 anti-discs large, RRID:AB_528203 | IF(1:50) |

*Appendix 1 Continued on next page*

*Appendix 1 Continued*

| Reagent type (species) or resource | Designation | Source or reference | Identifiers | Additional information |
|---|---|---|---|---|
| Antibody | anti-SMO (mouse monoclonal) | DSHB, *Lum et al., 2003* | DSHB Cat# Smoothened (20C6), RRID:AB_528472 | IF(1:100) |
| Antibody | anti-Rab7 (mouse monoclonal) | DSHB, University of Iowa, USA, | DSHB Cat# Rab7, RRID:AB_2722471 | IF(1:25) |
| Antibody | anti-FU (rabbit polyclonal) | Gift from *Ruel et al., 2003* | | IF(1:100) |
| Antibody | anti-β-Galactosidase (rabbit polyclonal) | MP Biomedicals | 085597-CF | IF(1:100) |
| Antibody | anti-CI (rat monoclonal 2A1) | Gift from *Motzny and Holmgren, 1995* | | IF(1:5) |
| Antibody | anti-mouse IgG Alexa Fluor Plus 555 (goat polyclonal) | Thermo Fisher Scientific | Thermo Fisher Scientific Cat# A32727, RRID:AB_2633276 | IF(1:200) |
| Antibody | anti-rat IgG Alexa Fluor Plus 647 (goat polyclonal) | Thermo Fisher Scientific | Thermo Fisher Scientific, Cat# A-21247, RRID: AB_141778 | IF(1:200) |
| Antibody | anti-rabbit IgG Alexa Fluor Plus 555 (goat polyclonal) | Thermo Fisher Scientific | Thermo Fisher Scientific, Cat# A32732, RRID: AB_2633281 | IF(1:200) |
| Antibody | anti-rabbit IgG Alexa Fluor Plus 488 (Goat polyclonal) | Thermo Fisher Scientific | Thermo Fisher Scientific, Cat# A32731, RRID: AB_2633280 | IF(1:200) |
| Antibody | anti-rabbit IgG Alexa Fluor Plus 647 (goat polyclonal) | Thermo Fisher Scientific | Thermo Fisher Scientific, Cat# A-21245, RRID: AB_253581 | IF(1:200) |
| Antibody | anti-mouse IgG cross adsorbed 488 (chicken polyclonal) | Thermo Fisher Scientific | Thermo Fisher Scientific, Cat# A-21200, RRID: AB_2535786 | IF(1:200) |
| Antibody | anti-mouse IgG Dylight 649 (donkey polyclonal) | Jackson Immuno | 715-495-151 | IF(1:200) |
| Recombinant DNA reagent (*D. melanogaster*) | BAC *smo* | BAC Resources PAC | CH322-98K24 | https://bacpacresources.org/home.htm |
| Recombinant DNA reagent | BAC *snap-smo* | This paper | | *snap* cDNA introduced in CH322-98K24 after Serine 33 codon of *smo* |
| Recombinant DNA reagent | *pENTR/D-TOPO-snap smo* | *Sanial et al., 2017* | | Template for *snap* amplification |
| Recombinant DNA reagent | *pENTR/D-TOPO-snap smo*$^{PKA-SD}$ | *Sanial et al., 2017* | | |
| Recombinant DNA reagent | *pENTR/D-TOPO-snap smo*$^{PKA-SD\ FU-SA}$ | *Sanial et al., 2017* | | |
| Recombinant DNA reagent | *pENTR/D-TOPO-snap smo*$^{PKA-SD\ FU-SD}$ | *Sanial et al., 2017* | | |
| Recombinant DNA reagent | *pUASt-GW-attB* | *Brigui et al., 2015* | | |
| Sequence-based reagent | rpsL-neo/smo mRNA 360 /F | Eurofins Genomics | PCR primers | AGGTTGCGATCTTATGCCTGTGGGTGGTCGCAGACGCATCGGCCAGTTCGggcctggtgatgatggcgggatcg |

*Appendix 1 Continued on next page*

*Appendix 1 Continued*

| Reagent type (species) or resource | Designation | Source or reference | Identifiers | Additional information |
|---|---|---|---|---|
| Sequence-based reagent | rpsL-neo/smo mRNA 462 /R | Eurofins Genomics | PCR primers | AGTTCCACATCCGACT GCTGCGCACTTGCGG GCGTTGTGCTGCCGA ACTTGGCtcagaagaact cgtcaagaaggcg |
| Sequence-based reagent | pEnSnapSmo/smo mRNA 360 /F | Eurofins Genomics | PCR primers | AGGTTGCGATCTTATG CCTGTGGGTGGTCGC AGACGCATCGGCCAGT TCG |
| Sequence-based reagent | pEnSnapSmo/smo mRNA 462 /R | Eurofins Genomics | PCR primers | AGTTCCACATCCGACT GCTGCGCACTTGCGG GCGTTGTGCTGCCG AACTTGGC |
| Sequence-based reagent | smo mRNA 289/Seq/F | Eurofins Genomics | PCR primers | GACTCGCCTCTGG CAAATGG |
| Sequence-based reagent | smo mRNA 512/ Seq/R | Eurofins Genomics | PCR primers | TGCCCTTCTTGGC GTACAGTCGG |
| Sequence-based reagent | Rpsl/neo/273/Seq/F | Eurofins Genomics | PCR primers | GAACTCCGCGCTG CGTAAAGTATG |
| Sequence-based reagent | snap/203/Seq/F | Eurofins Genomics | PCR primers | CCTACTTTCAC CAGCCTGAG |
| Chemical compound, drug | SNAP-Surface Alexa Fluor 488 | New England Biolabs | Cat#S9124S | 3.3 µM |
| Chemical compound, drug | SNAP-Surface Alexa Fluor 546 | New England Biolabs | Cat#S9132S | 3.3 µM |
| Chemical compound, drug | SNAP-Surface Block | New England Biolabs | Cat#S9143S | 2 µM |
| Chemical compound, drug | SNAP-Surface Alexa Fluor 647 | New England Biolabs | Cat#S9136S | 3.3 µM |
| Software, algorithm | Fiji | *Schindelin et al., 2012* | Fiji, RRID:SCR_002285 | http://fiji.sc/ |
| Software, algorithm | Adobe Photoshop | Adobe | RRID: SCR_014199 | https://www.adobe. com/products/ photoshop.html |
| Software, algorithm | Adobe Illustrator | Adobe | *RRID*:SCR_010279 | http://www.adobe. com/products/illustrator. html |
| Software, algorithm | Adobe InDesign | Adobe | RRID:SCR_021799 | https://www.adobe. com/fr/products/indesign. html |
| Software, algorithm | Prism – GraphPad 9 | Graph Pad Software | RRID:SCR_002798 | https://www.graphpad. com/scientific- software/prism/ |

