## [Editor Report]

In this paper, Gonçalves-Antunes and colleagues uncovered that the morphogen Hedgehog regulates the activity and subcellular localisation of Smoothened through vesicular trafficking. In particular, they demonstrated that Smoothened trafficking favours recycling and basal enrichment depends on its phosphorylation signature in a Hh-concentration-dependent manner. This work will interest a wide readership as it links Hh's functions as a morphogen with Smoothened's subcellular localisation.

---

## [Decision Letter]

[Editors' note: this paper was reviewed by Review Commons.]

---

## [Author Response]

General Statements

First of all, we would like to thank the reviewers for their insightful comments;

In summary, taking into account all the reviewers’ comments and advice we have now included a substantial amount of novel data which rely in a large part, on the construction of two novel genetic tools: a BAC SNAP-smo construct that allows expression of SNAP-SMO at endogenous levels and the first *fu* KO mutant (generated by CRISPR).

We now propose the following main changes in the Figures (the different sections of the text have been changed accordingly):

The Figure 1 has been completely reorganized.

(i) For simplification, we have removed the former panels A-A”, B-B”, E, E’ (data on Total SNAP-SMO) and E”. We have moved the data on Intra SNAP-SMO to the figure supplement 2.

(ii) We have added diagrams of the wing imaginal disc.

(iii) We now provide better images of SMO subcellular localization and apico-basal distribution thanks to an enhanced resolution confocal system: panels B-B” and C-C” that replace panels F-F” and G-G” respectively clearly show the basal localization of SMO in the “+HH cells “ (in the P and the CI-A region near the A/P).

(iii) We have kept the graph (former panel H) on the relative intensity in the four regions and added the graph on the relative intensity from the former sup Figure 1E.

(iv) We now show that the basal localization of SMO is suppressed when HH is inactivated (*hh^ts^* allele, panels F-F”).

It is now associated to three figure supplements.

Figure 1—figure supplement 1 shows that (i) SNAP-SMO carried by a BAC is fully functional and (ii) that overexpression of SNAP-SMO has no effect on HH signaling.Figure 1—figure supplement 2 includes (i) a schematic description of the labeling procedure (former Sup 1 F-F”) and of the quantification method (former Sup1 A), (ii) the data on Intra SNAP-SMO, and (iii) the validation of the inactivation the *hh^ts^* mutant at restrictive temperature.Figure 1—figure supplement 3 demonstrates that the HH-induced basal localization of Surf SNAP-SMO is not an artefact due to its fusion to the SNAP tag nor its overexpression. It shows the distribution of (i) immunolabeled endogenous SMO (former Figure S1 B-B” and C’ panels, the panel B has been removed for simplification as it did not provide novel informative data), (ii) Surf SNAP labelled SNAP-SMO expressed at endogenous levels from.

The Figure 2 is now associated to the Figure 2—figure supplement 1 that includes the former panels of Figure S2 and to which we added a control at 18°C (F) and experiments that show that blocking early endosomes (by the expression of RAB5^CA^) has the same effect on endogenous SMO or SNAP-SMO expressed at endogenous levels (G-G”’, H-H”’) that it has on overexpressed SNAP-SMO.

The Figure 3 is now associated to a sup Figure, Figure 3—figure supplement 1, in which we show that after the chase, Surf SNAP-SMO localizes with the vesicular marker RAB7 (A-A”’, B-B”’) and provide graphs in which we quantify the reduction of Surf SNAP-SMO during this chase(C-C’).

In Figure 4—figure supplement 1 (former S3), we have added (i) a graph showing the reproducibility of the distribution of Surf SNAP-SMO^WT^ in the Figures 1 and 4 and (ii) images showing the effects of mimicking or blocking phosphorylation of SMO on HH signaling.

The Figure 5 is now associated to two figure supplements:

Figure 5—figure supplement 1 includes the former S4 panels A-A’, B-B”, to which we have added a control XZ section showing GFP-FU without the GRAB tether.Figure 5—figure supplement 2 includes the former S4 panels C-C”’, D-D”’ and E-E’ to which we added data that shows that apical grabbing of GFP-FU leads to an increase in Surf SNAP-SMO (expressed at endogenous levels from a BAC).

In Figure 6, for simplification we have removed the data on CI-F and on *dpp*. We have also added data showing that tethering FU to the apical region is not suppressed in the absence of endogenous FU (CRISPR KO mutant of fu).

It is now associated to a figure supplement (Figure 6—figure supplement 1) showing the effects of the Nvr1 and T48 tethering of GFP-FU on *dpp* and a scheme on the construction of the *fu ^KO^* mutant.

In Figure 7, the model has been simplified.

**Responses to reviewer #1:**
Specific points:1. The phenotypes under study are subtle – we're talking about a shift of a few percent of Smo from the most apical region to basal. The work appears to have been carefully done, though, with sufficient replicates to allow these small differences to be detected using statistics. However, many of the conclusions are based on a subsequent comparison of the differences of differences (e.g. "a reduction of 46% in the far anterior cells but of only 25% in the posterior cells" p. 9). Given the magnitude of the uncertainty (standard deviation) in the individual measurements, it's not always clear that these differences between the differences are really significant, and this is not tested statistically. In light of the subtlety of some of the effects, applying a statistical analysis to these comparisons would strengthen the conclusions.

We have now statistically compared the % of decrease in Surf SNAP-SMO between t0 and t15 (calculated as (t0-t15)/t0) in the FA (-HH) and P regions (+HH). This confirms the significance of these comparison. This comparison is shown in the Figure 3—figure supplement 1C, C’.

Concerning the shi^ts^ experiments, we removed the part on comparing the % (lines 147 to 153) as it was unnecessary long, and the effects presented in the graph C were sufficient and statistically significant.

2. The images in Figure 1 show a pronounced enrichment of total SNAP-Smo in the apical domain of both far A and P cells, which is not seen when staining for endogenous Smo (shown in Supplement). This could give a bit of a misleading impression about the "normal" distribution of Smo, which is more uniform. Can the authors comment on this?

We have now built a transgenic line carrying a *smo* BAC construct in which we inserted a SNAP tag at the exact same position than in the *UAS SNAP-smo* construct. We show that this construction is fully functional (now shown in Figure 1—figure supplement 1 A, B-B”, see also text lines 114-116). Labelling of Surf SNAP-SMO expressed in this condition is shown in Figure 1—figure supplement 3C-C”, see also text lines 158-159. No strong apical enrichment is seen. As SNAP-SMO is initially targeted to the apical membrane before being endocytosed, a possibility is that apical endocytosis might be a limiting step when high amounts of SNAP-SMO are synthetized, leading to its apical accumulation. However, SNAP-SMO expressed for the BAC at endogenous levels is -as the case for endogenous SMO and overexpressed Surf-SNAP-SMO-, enriched in the basolateral region of the cells in the Ci-A and P regions, which is the main subject of this work. Note that we could not correctly quantify these images due to a low signal to noise ratio and strong photobleaching, which prevented us from doing Z stacks.

We now also show that endogenous SMO and SNAP-SMO expressed from the BAC at endogenous levels are also initially targeted to the apical region where they are endocytosed in a RAB5 dependent manner, indicating that (SNAP-SMO overexpression does not affect its apical targeting nor its apical endocytosis). This is now shown in Figure 2—figure supplement 1G-G”, H-H”. See also the text lines 201-204.

Altogether these results strongly validate the use of this UAS *SNAP-smo* construct.

See also the response to reviewer 2, point 1 p8.

3. Although everything is quantified, in a couple of cases the quantification doesn't seem to match the eye test based on what is shown in the images, or the staining is inconsistent from figure to figure. Can the authors comment on the following points?3A. The graph in S1D' shows no difference in intracellular SNAP-Smo mean intensity distribution between FA and P cells. In the matching image in Figure 1D', it looks a lot like intracellular Smo levels are much lower in P and Hh-responding A cells than in far A cells – both in the number of Smo+ punctae and in their staining intensity.

We thank the reviewer for this comment. We apologize for what is an error in the images frame and we now provide the images with a larger section of the P region and with the Ci staining that allows identification of the different regions. See Figure 1—figure supplement 2D,D’.

3B. The differences in surface SNAP-Smo distribution in far A (apical dot like structures, presumably endocytic vesicles) versus P cells (on membranes, fewer vesicles) shown in Fig, 1C and Dare not consistent in other staining’s of the same genotype (e.g. Figure 2A, Figure 4A).

Some differences reflect the biological variation from one disc to another, which justifies the importance of systematic quantification. In this case, the estimated average intensity of SNAP-SMO labeling was determined after quantification of Z projections, each composed of eight sections, from between 10 and 23 discs and provides more robust information than a single image.

Moreover, we have now also compared the apico-basal distribution of SNAP-SMO in the experiments quantified for the Figures 1 and 4 and this confirms that the apico-basal distribution of SNAP-SMO is in fact very reproducible. We have added this data as Figure 4—figure supplement 1E.

Note also that some comparisons are not equivalent, such as the comparison between (i) Z projection and Z sections, (ii) experiments done at different temperature (Figure 2 compared to Figures 1, 3 and 4), and now (iii) experiment done with different confocals as the confocal Zeiss LSM980 spectral Airyscan (63X, reconstructed Z sections) in the novel (Figure 1) and the confocal SP5 AOBS images (40X, direct Z imaging) in almost all the other Figures.

4. The experiments comparing apical versus basolateral trapping of FU are a bit complicated to interpret. What is the evidence that NVR1 leads to basolateral enrichment of FU? Put another way, how does FU localization when co-expressed with NVR1 compare to its distribution when just expressing FU alone?

We have now added in an image of GFP-FU in Figure 5—figure supplement 1C. Comparison with the Figure 5B’, D’ and Figure 5—figure supplement 1C’, D’ confirms that GFP-FU is indeed relocalized by the Nvr1 and T-48 traps, respectively. See text lines 285-286.

5. One caveat to this experiment is that apical targeting of FU will presumably lead to a much higher FU density since the apical domain is fairly small. Spreading the same amount of FU over the entire basolateral domain will result in much less concentration of the protein. Since Fu clustering plays an important role in its activation, can the authors rule out that the basolateral trapping doesn't have much effect because Fu activity is much lower in this condition?

We cannot formally exclude the possibility raised here to explain the ectopic *ptc* expression (but not the reduced *en* expression). However, to our knowledge, there is no data indicating that increased levels of FU could promote its dimerization nor its activation (on the contrary, overexpression of FU has no effect on HH signaling for instance see Claret et al.). On the contrary, FU levels are decreased in response to HH (Ruel et al., 2013). Moreover, an effect via FU clustering would not explain the effect of T48-anchored FU on SMO localization as (Shi et al., 2011) also showed that dimerized FU activates CI independently of SMO.

Note also that many labs (Kalderon, Jian, our …) have overexpressed FU, with or without a tag, even at very high levels, and activation of FU or the pathway has not been observed.

Even if this apical tethering would act indirectly by increasing FU clustering, this would not change the fact that it allows FU-GFP to activate the medium to low HH-targets, but not high levels of the anterior *en*. This also indicates that a secondary event is necessary for the highest level of activation.

We did not see the points 6 and 7.

8. The conclusion that a basolateral function of FU is missing in the apical targeting experiment would be clear if the experiment was done in a fu mutant background. However, in wild-type cells, shouldn't there be the usual abundance of endogenous FU to carry out the function of FU in the basolateral domain?

We have now repeated this experiment in a *fu* null mutant background (thanks to *a fu^KO^* mutant that we built by the CRISPR method). In this context, apical trapping of GFP-FU has similar effect in the absence as in presence of the endogenous FU protein. They indicate that apically trapped GFP-FU does not act via endogenous FU and reinforce our conclusion that FU is required basally for the full activation of High HH targets. Note that we also checked that GFP-FU rescues the effects of the fu^KO^ allele. These data are presented in Figure 6E-H. See also the text lines 316-332.

Minor commentsp 9 – It's not clear to me why the authors conclude that endocytosis of Smo occurs all along the apical-basal axis in Figure The data in E suggest that endocytosis is primarily occurring apically.

Our data show that a strong block of endocytosis (Rab5 CA) leads to an accumulation of SMO in the apical region almost exclusively, revealing that SMO is initially targeted to the apical region where it undergoes a first endocytosis. However, partial blocking of SMO endocytosis (under shi inactivation conditions) leads to an increased accumulation of SMO in both the apical and the basolateral region, which reveals that SMO redistributed to the basolateral region can undergoes a secondary endocytosis.

Figure S4E – What condition is being analyzed? I couldn't find it in the legend.

We apologized for this lack of information. This point has now been added in the legend of this Figure now called Figure 5—figure supplement 2.

SignificanceThere have long been bits and pieces of data about Smo cycling to the plasma membrane and being internalized, and Ptc affecting this process to keep Smo levels low in the absence of Hh. However, there is no clear picture of the route that Smo takes in response to Hh, at least in flies. (In mammals there is good evidence that Smo goes to cilia upon its activation.) The authors have used an innovative approach to try to address this, by fluorescently labeling surface Smo and trying to follow its fate. While subtle, the conclusions that Smo is delivered to the apical domain, endocytosed, and in response to Hh moves basally are generally convincing, with some exceptions noted above. The data showing that this is controlled by PKA/CKI and FU phosphorylation of Smo are also clear. Aside from these mechanistic experiments, the rest is more descriptive and, as the authors note, the results don't clearly distinguish whether the basal localization of Smo is a consequence rather than a cause of Smo activity. Nonetheless, it's a quality manuscript that will be of interest to many in the Hh field, and likely to people who are more generally interested in GPCR signaling.

Concerning the distinction between a consequence or a cause, we formally agree. We have tried to test the “cause hypothesis” by using the GRAB system to target mutant and wild-type forms of SMO-GFP. However, we did not succeed in relocalizing SMO as on the contrary, we observed that SMO-GFP had the ability to relocalize the anchors T48 (Author response image 1) and NVR1. We therefore see no other possibility to discriminate between these two hypotheses.

**Author response image 1. sa2fig1:** 

We would however like to mention that while numerous published data show a strong correlation between SMO cell surface localization and SMO activation (i. e. (i) HH leads to both SMO activation and its relocalisation at the cell surface , (ii) SMO relocalisation to the cell surface is sufficient to activate SMO and (iii) blocking SMO phosphorylation by the PKA prevents both its localization at the cell surface and its activation) none of them- to our knowledge- show that this relocalisation is in fact required for its activation. Similarly, all our results strongly connect SMO basal localization to its “high activation” without formally proving that the second requires the first as: (i) high HH leads to SMO basal localization, (ii) the phosphorylation of SMO by FU is required for its basal accumulation, (iii) blocking FU in the apical region reduces both basal localization of SMO and its high activation. We have now added data (see Figure 4—figure supplement 1 F-F”, G-G”, lines 266-271) confirming that the phosphorylation of SMO by FU is required for expression of the high HH target gene *en* and some text in the discussion lines 390-394. This further links the connection between SMO basal localization and activation. See also our response to reviewer 3 p19.We want to stress that the significance of our data also has to be evaluated in light of what is known on the HH reception: First, HH is known to form a basal gradient supported by cytonemes and this basal gradient is necessary for high HH signaling (Bischoff et al., 2013). Second, the HH co-receptor PTC, is present both in the apical and basolateral/basal regions and was recently shown to inhibit SMO by removing accessible cholesterol from the outer leaflet of the plasma membrane, with HH binding to PTC blocking the passage of these cholesterol molecules. As the apical and the basolateral membranes behave as distinct membrane regions, with no exchanges of lipids and proteins; the basal molecules of PTC that receive HH from the basal gradient should need basal SMO to fully activate the pathway. Given all this and our data, we propose that FU would be required to send SMO to the basal region; due to the high levels of HH in that region, SMO would then be fully activated, leading to an increased activation of FU, which would then inhibit its partners and targets COS2 and SUFU to fully activate CI.

*Responses to Reviewer #2*:

Major comments:1. While SNAP labeling technique is a powerful method for probing protein localization, the main question of this manuscript is whether overexpressed SNAP-Smo reports the true localization pattern of endogenous Smo. As shown in Figures 5B" and S1B as well as in the published literature, endogenous Smo does not show significant apical accumulation but is more basolateral, which is very different from the SNAP-Smo shown in Figure 1B. The authors claimed that SNAP-Smo is functional in vivo by citing Sanial et al. (2017) that overexpressed SNAP-Smo rescues adult wing defects caused by smo RNAi. However, Sanial et al. (2017) did not completely rule out whether an N-terminal SNAP tag would lead to Smo activation, or whether overexpressed SNAP-Smo represented an activated form of Smo. If this is the case, the studies described in this manuscript are less important for understanding the impact of the subcellular localization of Smo in Hh signaling activation. Furthermore, the dimerization requirement for Smo activation further complicates the situation. To vigorously establish the relevance of polarized SNAP-Smo distribution in Hh signaling, it is especially important to express SNAP-Smo at levels comparable to endogenous Smo or knock in the SNAP tag into the smo locus.

The effect of the tagging of SMO with the SNAP-tag and of SNAP-SMO overexpression on SMO apico-basal distribution is addressed in our responses to reviewer 1 point 2. In summary, using a recombinant BAC carrying a SNAP-SMO fusion (expressed under the endogenous promoter of *smo*) we show that, similarly to what is seen with overexpressed SNAP-SMO, SNAP-SMO expressed at endogenous levels (i) fully rescues a *smo* loss of function (viability, wild-type wings and targets, in Figure1—figure supplement 1A, B-B”, (see text, lines 113-116) and (ii) is initially addressed apically where it is endocytosed in RAB5 endosomes Figure2-supplement 1H-H”, text lines 202-204).

Concerning the possibility that SNAP-SMO overexpression would promote its activation, we provide new data, which shows that overexpression of SNAP-SMO has no effect on *en* expression nor the presence of the CI-A domain, two markers of high HH signaling. These results are now shown in Figure1—figure supplement 1C-C” (text lines 115-116) and confirm publications by numerous labs (Beachy, Kalderon, Thérond, Jia, Jiang, our lab etc…) showing that overexpression of SMO is not sufficient to activate it. These reports are based on the overexpression of various UAS *smo* constructs, untagged or with various tags inserted in various loci (N term or C term).

2. The authors claimed that newly synthesized SNAP-SMO proteins display "dot-like structures" when released from the endocytic block, likely corresponding to endocytic vesicles. This claim is untenable unless co-staining is performed with endocytic markers.

We now show colocalisation between some of these *Surf SNAP-SMO* “*dot-like structures" and* the endosomal marker RAB7. This is presented in Figure 3—figure supplement 1 A-A”, B-B”. See also the text lines 223-224.

Furthermore, there are no apparent differences between the mean intensities shown in Figures 3A and 3B. In some cases (lateral and basal), the mean intensity of the surf SNAP-SMO even increased after the 15-minunte chase, which is not consistent with the statistical analysis in Figure 3C.

As we now add in the Figure 3 legend, the discs were treated in the same conditions and all the images were acquired under the same conditions. However, the dynamic range was automatically adjusted for each image independently by Image J for a better visualization of the subcellular localization of SNAP-SMO. To allow a better eye comparison, we have now replaced these images by images in which the dynamic range is normalized.

3. The XZ images of the apical-basal distribution of SNAP-Smo and endogenous Smo shown in different figures are inconsistent. For example, surf SNAP-Smo accumulated apically in Figures 1D, 1G and Figure 2D, but not in Figures 2A and Figure 4A.

As explained in our response to reviewer 1 point 2, only the images 1D and 4A can be compared for Surf SNAP-SMO and we have now added in Figure 4—figure supplement 1E data showing that both sets of experiments display very similar distribution along the apico-basal axis both without and with HH, even if differences in the contrast of the images, may be misleading for the eye.

Furthermore, in Figure 5B', significant accumulation of endogenous Smo was observed when overexpressed Fu was captured in the apical region. However, the upregulation and basal accumulation of endogenous Smo in Figure S4C' were relatively weak under the same conditions. Although immunostaining results for SNAP-Smo or endogenous Smo may vary, it is necessary to provide more appropriate results consistent with statistical data.

We believe that there is a misunderstanding as in Figure 5B’, GFP-FU is shown, not SMO and it is enriched in the apical region due to its anchoring by T48. Endogenous SMO, which is shown in Figure 5B”, shows no apical enrichment. As mentioned, above (response to reviewer 1, point 4) we have added as a control the subcellular localization of GFP-FU in the absence of an anchor Figure 5-supplement 1C (text line 285-286). Note also that Figure 5 shows XZ sections in the anterior region (-HH situation, corresponding to the FA region); while in the previous Figure S4 (now Figure 5—figure supplement 1) the XZ sections are in the posterior region (+HH situation): this explains why the effect are weak in the P compartment where SMO is already stabilized and activated.

4. Capture of overexpressed GFP-Fu in the apical region resulted in a marked increase in endogenous Smo proteins, which accumulated slightly in the basal region, and downregulation of high Hh signaling targets, such as en. In contrast, capturing overexpressed GFP-Fu in the basolateral region had no apparent effects on Smo levels or apical-basal distribution. The authors concluded that apical-captured GFP-Fu activates and stabilizes Smo on the cell surface, while basal Fu is required for basal enrichment of Smo and activation of high-level Hh targets. However, previous studies have shown that Fu can dimerize. Therefore, the captured GFP-Fu may indirectly affect the localization of endogenous Fu, thereby affecting its function on Hh target expression. The amount and apical-basal distribution of endogenous Fu should be examined to eliminate this possibility.

As mentioned in our response to reviewer 1, point 8, we now show that the effects of the apically trapped GFP-FU on HH signaling do not depend on the presence of the endogenous FU protein.

Note also that we have no way to assess the localization of the endogenous FU due the lack of a tool that would allow us to distinguish endogenous FU from overexpressed GFP-FU.

5. Since this study highlights the effect of Hh and Fu on Smo protein levels and their apical-basal distribution, it is necessary to examine whether reduced expression of hh or fu affects the amount and polarized distribution of endogenous Smo.

To ensure that the changes in the apico-basal distribution of Surf SNAP-SMO were indeed due to HH, we performed the same experiment when HH function was inactivated, using a thermosensitive allele of *hh*. Under HH inactivation, Surf SNAP-SMO is no longer accumulated in the P and the anterior cells near the A/P, and is no longer enriched in the basolateral region of these cells. We have added this result in Figure 1F-F” and Figure 1—figure supplement 2F-F” (text lines 162-169).

Minor comments:1. In the Discussion, the authors stated that aPKC, a kinase previously known to regulate polarized distribution of Smo, does not affect the "high Hh"-dependent basal localization of Smo. However, no relevant data are provided in the paper and no references are cited.

We apologize for this confusion: we meant that the aPKC was a kinase previously known to regulate polarized distribution of SMO but that it is not known whether the loss of aPKC affects the "high Hh"-dependent basal localization of Smo. We have changed the text accordingly to make it accurate (lines 368-370) The reference is Jiang et al., PNAS 2014 and has been added (line 368).

2. Overexpression of the captured GFP-Fu protein in the dorsal compartment should have no effects on Ci-FL levels in cells localized in the ventral compartment. However, in Figure 6D, Ci-FL levels and activation status (i.e. CiA) in the ventral compartment appear to be altered by overexpression of captured GFP-Fu in the dorsal compartment.

In Author response image 2, mostly the dorsal part of the wing pouch is shown. We apologize for that and you can see the full wing pouch in the image to the right. However, on the revised version of the MS to simplify this part we have decided to delete these data.

Nature and significance of the advance (e.g. conceptual, technical, clinical) for the field.The idea is new, but the evidence in the current manuscript is not convincing.

We hope that the additional data that we now provide with more control experiments and which include results with the BAC and the *fu^KO^* mutant make the novel manuscript convincing. See also our above response to the Reviewer 1 “Significance” section.

Context of the existing literature (provide references, where appropriate).

Li et al. (2016) PLoS Biol. 14, e1002481.

Li et al. (2018) Sci. Signal. 11, eaan8660.

Malpel et al. (2007) Dev. Biol. 303, 121.

Sanial et al. (2017). Development. dev.144782.

Shi et al. (2013). J. Biol. Chem. 288, 12605.

– Audience that might be interested in and influenced by the reported findings.

Developmental biologists and cell biologists who are interested in mechanisms of signal transduction

– Field of expertise with a few keywords to help the authors contextualize your point of view. Indicate if there are any parts of the paper that you do not have sufficient expertise to evaluate.

Development, signal transduction, endocytic transport, polarized protein distribution

**Responses to reviewer #3:**
Key findings:HH signal causes more SMO protein to be central and basal in wing imaginal disc cells, with reduction apically.Blocking endocytosis in two ways causes SMO accumulation in apical regions of cells. HH does not appear to affect SMO endocytosis.Pulse-chase labeling showed movement of SMO from apical to more central and basal regions. Since this looks HH-independent, the authors conclude that HH regulates SMO abundance by altering degradation rather than endocytosis.SMO altered to prevent cytoplasmic tail phosphorylation, or to mimic it, shows that apical-basal SMO localization is regulated by phosphorylation.The most striking result in the paper is that apically localized over-produced FU kinase causes SMO to become active with respect to medium (not high)-level HH target gene activation and to accumulate at cell membranes.Lines 351-2: "we cannot exclude that this basal [SMO] localization is a consequence rather than a cause of SMO activation".

See our response to reviewer 1 “significance” section and in the discussion lines 390-394.

The authors propose that basal localization driven by high HH brings SMO into a membrane environment where it gains activity and activates high-HH target genes.Line 24: "plasma membrane subcompartmentalisation". While that may be correct, most of the data do not discriminate between plasma membrane and internal membranes. This makes it harder to make a model, such as the one in Figure 7.

We have deleted “plasma membrane subcompartmentalisation” from our summary.

Specific comments:line 47 There are reviews more current about Hh and cancer therapies than the one mentioned, Briscoe & Therond 2013.

We have replaced this reference by Ingham PW. Curr Top Dev Biol. 2022 Line 47.

line 67 "cytotail" is jargon, not English, and should be replaced.

We have replaced it by “cytoplasmic C-terminal tail”everywhere.

line 99 "phosphomutants" is jargon, not English, and should be replaced.

We have replaced it by “phosphorylation mutant”.

Figure 1 would be helped by having a diagram showing an imaginal disc and the regions within it referred to in the text. This would support Figure 7 as well. Figure S1A is closer but still not a full-context view.

Such diagrams are now shown in Figure 1A and A’.

Also in Figure 1, DLG is used to stain. A search of the text reveals that the first mention of DLG is in line 405 in the methods, and what it stains is only mentioned at line 440. The use of it should be explained in text and legend with reference to Figure 1.

We thank the reviewer for noting that and this has been done. See line 134.

lines 118-120 Shouldn't there be a third category of cells near the A-P boundary that receive HH but do not produce it? This would be, I suppose, the region indicated by the purple A in Figure S1. This question is especially relevant since it is unclear whether any of the cells shown in Figure 1A-D represent the FA region-there is no labeling to tell us. And I see that in line 285 these cells are finally mentioned.

As mentioned at the beginning of this document, and following this very useful comment and the following suggestion to simplify Figure1, we have now totally reorganized this Figure, its three figure supplements and the corresponding text section. We now directly introduce the presentation of these different regions along the A/P axis.

line 122 Yes, stronger apical accumulation of SMO is visible in A, B, C, and D but…line 123 I fail to see the increased basal accumulation; indeed, I can scarcely see any basal accumulation.

We now provide -in the novel Figure 1 XY and XZ- images taken with a Zeiss LSM980 confocal with a spectral Airyscan 2, 63X (rather than with a Leica confocal SP5 AOBS, 40X). These novel images clearly show the basal accumulation of Surf SNAP-SMO in presence of HH. However, given that the confocal Zeiss LSM980 spectral Airyscan acquisitions are lengthier and do not allow us to perform direct XZ sections, all the quantifications and most of the other images still correspond to the SP5 images (except for the Figure 5—figure supplement 1B-B”).

I do not know how to reconcile the A-D panels showing staining with the chart in E, which shows the 50% increase in P vs FA regions. Is that because the FA regions are not shown in A-D? E' and E' show scarcely any staining in basal regions, which is what I see in the stained discs. Perhaps the use of a basal marker, analogous to the use of DLG, would clarify the situation. Figure S1A at least has things labeled more clearly, and there too barely any staining is seen (A or P) in the basal region. In Figure S1 the anterior is divided into three regions depending on Ci staining (not shown) with the red R having an arrow to "FA". Yet the red R region is not very far anterior.Overall, therefore, Figure 1 is quite confusing and does not seem to fully agree with the conclusions mentioned in the text. The references in the text to FA and P are not matched by any labels in Figure 1A-D. What I do see, in A, B, C, and D, is higher levels of surface SMO in lateral regions in the posterior vs anterior-and maybe the same for total SMO though A seems to contradict B in that regard. So at line 105, the accurate statement would seem to be "lateral" rather than "basolateral", and it is not clear that the protein being monitored is only cell surface protein. Part of the problem may be the lack of indication in the figure of what counts as "basal".

We thank the reviewer for this advice, which prompted us to reorganize and improve this Figure 1 and its associates figure supplements 1, 2 and 3, as we described at the beginning of this document.

The mentioned graph E does not address the apico-basal localization of SMO but shows the increase of SMO levels in response to HH. But, following the advice to simplify Figure 1 and given that it mainly confirmed a fact that has been well documented by many labs (including our), this part has been removed from our revised manuscript.

As mentioned in the text (former line 116, now in line 135) and in the former Figure S1A., we call “basal” the “10% most basal region of the Z sections”. To our knowledge, the basal region could be not defined by using a marker as we can do for the apical region.

The use of DLG and the definition of what we call basal are now clearly explained in the main text (line 134) as well as in the legends of Figure 1—figure supplement 2.

We hope that the confocal Zeiss LSM980 spectral Airyscan images convincingly display the basal accumulation of Surf SNAP-SMO in the presence of HH.

See for more details our response to reviewer 1 point 3, p 4.

Figure 1E' shows two sets of ***, which may be correct statistically but seem like small differences.

Despite the identification of many genes and proteins involved in HH signaling, little is really known on how the differences in HH dose and even less on how the basal and apical gradients are sensed and interpreted. Classically, such “all or none “responses may implicate regulatory loops that could contribute to “switch like “effects (Ashe et Briscoe Dev. 2006). In that context, quantitative imaging approaches as the one we developed, could allow -when associated to statistics to validate them- to identify effects that may be small but very relevant when considered in the context of the spatiotemporal regulation of a complex system. See also below.

line 119 Not really a model if they are HH-responsive cells.

We understand this point but we think that we should keep this term because strictly speaking the P cells do not really “respond to HH” in the way A cells near the A/P do: in the latter case, SMO is activated because the binding of HH to its receptor PTC inhibits the negative effect exerted by the later on SMO, while in the P cells, SMO is activated because PTC is absent (the *ptc* gene is not expressed in the P compartment).

line 123 "an increase"

This has been changed

line 129 "integraded" should be "integrated"

We thank the reviewer for noting these spelling errors and we have made these corrections.

lines 131-3 if it's not significant, why mention it?

We have now removed this in the result section and line 352 in the Discussion section.

line 135 in S1B, B' it is again unclear what is meant by basal, since the only staining seems to be lateral.

See above.

In S1C the key shows apical as light green, but nothing in the chart is light green. In contrast green bars in Figure 3D do not have a corresponding item in the key. The higher level of SMO in the posterior is visible. "Column" is defined in the text at line 130 but should be repeated in the S1 legend.

We apologize for these mistakes in the keys and we thank the reviewer for noticing it.

The colors of the graph have been changed on Figure3D. For simplification purpose, the panel S1C has been removed (see Figure1—figure supplement 3B), showing the images and the chart with the relative intensities.

We now mention that “Column corresponds to the entire height of the epithelium (i. e.100%)” in the legend of the current Figure1—figure supplement 2.

I can't see the basis in the staining for the right hand two (dark blue) bars in Figure S1C, since in B and B" the SMO staining seems weaker apically and laterally as well as basally in what may be FA cells vs P. That's why C' seems a better read of what's going on, i.e. no change in FA vs P in relative terms in any of the three regions (this is acknowledged in lines 150-2 but with different emphasis).

We have followed this advice and removed this panel.

lines 154-6 and Figure 1C' and D': Since the signal for intracellular Smo is so weak, I think the conclusion that HH does not affect it is also weak.

We show this data as the signal is detectable, was carefully quantified, and statistically analyzed. After organization of the Figure 1 and taking into account the present comment, this part has been moved to the Figure1—figure supplement 2.

lines 167-8 These too are small effects but supported by the pattern across four regions.lines 168-9 Not sure what point is being made here. First of all both A and P compartments are being used here, and second the disc is a long-established tool for studying HH signaling.

See above our response to the comment.

lines 180-1 The proper control here (for Figure 2A) would be the shibire genotype without the temp shock, Figure 2E notwithstanding.

This control is now added to the Figure 2—figure supplement 1E and is now mentioned in the text line 190-191.

lines 204-210 If much of the normal signal is endocytosed SMO, then describing its location as "lateral" in the earlier paragraphs and figures is a bit misleading. Lateral suggests the sides of cells, not the inside. "Central" might be a better term.

We agree that after the chase we cannot distinguish between the labeled SNAP-SMO molecules which are still at the cell surface from the molecules that have been endocytosed. We have therefore changed “lateral” to “intermediate” in the Figure 3 (and the corresponding text lines 233, 236, 239 and 241).

lines 218-9 nice clear resultlines 228-9 Small differences but okline 232 "transcytosed" would mean traversing the cytoplasm to appear at the other (in this case basal) surface. Instead what's observed is an apparently internal central (and/or lateral) accumulation.

We agree that we cannot not conclude this in this section. This term was removed.

line 239 "cytotail" is not English

We have replaced it by “cytoplasmic C-terminal tail”.

lines 256-7 if the "basolateral" enrichment is a sign of the activation of SMO, why does a primarily negative regulator like PKA have an effect similar to the positive regulator HH? In contrast Fu is primarily a positive regulator.

Indeed, PKA was identified as a negative regulator of HH signaling in *Drosophila* as it negatively regulates CI. However, it has been shown by many labs (Jia, Jiang, Zhu, Beachy etc…) that it also acts as a positive regulator of SMO. To clarify this, we have changed “Given the importance of the phosphorylations of the C-terminal cytoplasmic domain of SMO” by “Given that the PKA and FU kinases positively regulate SMO activation and accumulation at the membrane” See lines 247-248.

Line 29-5 FU-GFP (T48) is enriched apically, but is also present throughout the cell. FU-GFP (NVR1) is not enriched apically but is seemingly present throughout the cells. Therefore interpretation of any negative result (eg Figure 5C' and lines 290-2) depends on whether the amount of FU kinase is limiting or not.

Several publications reported that changes in *fu* dosage (either loss of one dose of the *fu* gene (see for instance Preat et al. 1990)) or its overexpression (see for instance Claret et al., 2007) have no effect. Moreover, we now show that the effects of apical trapping of GFP-FU on *hh* targets still occur in the absence of the endogenous FU protein. All this indicate that is not likely due to an effect on the amount of FU kinase. See Figure 6 E-H and lines 316-332. See also our response to reviewer1, point 5, p5.

line 286-8 Nice result.line 288 explain what the normal "anterior" expression of en looks like-it's not everywhere.

Normal anterior *en* expression (also called anterior *en* or late *en* expression) occurs in the first rows (around three) of cells abutting the A/P boundary. Note that *en* is also expressed in the whole posterior compartment but this is independent of HH. We have added this information in the legend of the Figure 1—figure supplement1.

line 297 "enrichment"

This has been corrected.

line 294 the reference to "cell surface" came as a surprise, since the surface means apical, lateral, and basal. Figure 6 doesn't resolve surfaces and Figures5A', B', C', and D' do not clearly show what is happening to SMO localization with respect to cell surfaces.

We have been able to visualize Surf SNAP-SMO expressed at endogenous levels from the BAC construct. It shows expression of GFP-FU with T48 leads to accumulation of Surf SNAP-SMO. This result is shown in Figure 5—figure supplement 2C-C”’. Lines 299-301.

line 298 Logic not clearly explained here. For example, why can't the endogenous FU provide this proposed basal function?

We think that apically tethered GFP-FU recruits endogenous FU. This is in agreement with the fact that the effect of apically trapped GFP-FU does not require endogenous FU. See also our response above and to reviewer 1 point 8, p5.

line 632 "merged" not "merge"

This has been corrected

Figure 7 This diagram seems unnecessarily complex to me and should be simplified as much as possible.One problem is that the design of the diagram makes it hard to see quantitive changes in SMO abundance in different regions of the cells, yet those changes are central to the paper.How much the abundance changes occur inside cells vs on the plasma membrane is not clear, and the diagram should leave open both possibilities.Why are PKA/CKI excluded from the right panel? I don't think that including adherens junctions, integrins, and extracellular matrix does anything but make the model harder to understand.Golgi appears in the model but nowhere else in the paper, so it's not clear why it's brought in here.

We thank the reviewer for these suggestions. The diagram is now simplified and PKA/CKI added.

SignificanceHedgehog (Hh) signaling is employed in the shaping of many organs and tissues during the development of many species of animals. The mechanisms of Hedgehog signaling are important for understanding development, birth defects, evolution, and cancer. In this paper the authors examine mechanisms involving the transmembrane protein Smoothened (SMO), a positive regulator in the pathway from received HH signal to the activation of target gene transcription. Subcellular localization of SMO has been shown to be important to its activity state. In vertebrate cells, arrival of a HH signal causes SMO to accumulate on the surface membrane in primary cilia, one of which is present on the surface of every cell. No analogous structure has been discovered in the *Drosophila* cells used in the present study, but SMO subcellular localization has been found to vary in response to HH, including some of it moving to the cell surface. For example, the present authors found that a positive feedback loop between the FU kinase and SMO increase the accumulation of both proteins at the cell surface.Hh signaling was discovered in *Drosophila* and found to be important both during embryonic development and in the imaginal discs, precursors of adult appendages and body wall. Here the authors analyze effects on SMO during Hh signaling in the wing imaginal disc. They find that HH causes SMO to become stabilized during steps of endocytosis and recycling, and that at the highest levels of HH signaling the SMO protein is enriched in basal domains of epithelial cells.The authors have done a lot of precise and difficult immunostaining work to look at the localization of SMO. The results show often small changes (10% effects) that are statistically significant but leave open questions of how essential the subcellular localization of SMO is to its functions.

We believe that the answer to these points relies on the complexity of the system that we study and on the fact that in “real life” populations of signaling molecules are both heterogeneous and dynamic.

First, one has to have in mind that in response to HH, only a part of the SMO population is phosphorylated by the PKA/CKI at a given time and among those, a fraction is also phosphorylated by FU as well (see for instance Sanial et al. 2017). Thus, only a fraction of it is expected to be localized in the basal region. In that respect, the use of mutants that mimic or block its phosphorylation reduce this heterogeneity, making the effect of the phosphorylation more visible and allowing the dissection of the role of the phosphorylation (see Figure 4). For instance, SMO^PKA-SD FU-SD^, which mimics highly phosphorylated SMO, is more enriched basally than unphosphorylated SMO (SMO^WT^ in absence of HH) and this effect is reduced when the Fu sites are mutated to prevent their phosphorylation. Of note, we now have included data showing how these mutations of SMO also promote its activity. It shows that SMO^PKA-SD FU-SD^ can promote high HH signaling, while SMO^PKA-SD FU-SA^ on the contrary, blocks it. All these results further strengthen the link between the basal localization and its signaling activity. See the Figure 4—figure supplement 1 F-F”, G-G” and our response to reviewer 1, p 6-7.

Second, these effects on SMO trafficking are likely very dynamic and we may have “caught” a transient basal localization of SMO. For instance, the “very active” basal fraction of SMO may undergo subsequent degradation leading to its desensitization.

Finally, the response to HH levels, may also require the additive, and even possibly synergetic effects of other regulatory loops as those that have been shown to regulate HH signaling (see for instance Strigini and Cohen 1997, Kent et al., 2006, Holmgren 2022).

See also our response to reviewer 1’s comment, “significance“ section, p6.

This paper is part of the ongoing effort in many labs to understand the molecular biology of HH signal transduction. It will therefore be of interest to others working in this arena, and has implications for developmental biology and cancer. The most striking advance in this paper is the importance of FU kinase subcellular localization and its impact on SMO function and HH target gene activation. The paper could be simplified and made more accessible by focusing it more on these results. Localization of SMO and other transducers has proven to be very important for understanding HH signaling in flies and mammals, so this paper provides some useful new ideas.